# CompressKV: Semantic Retrieval Heads Know What Tokens are Not Important Before Generation

## Abstract

Recent advances in large language models (LLMs) have significantly boosted long-context processing. However, the increasing key-value (KV) cache size poses critical challenges to memory and execution efficiency. Most KV cache compression methods rely on heuristic token eviction using all attention heads in Grouped Query Attention (GQA)-based LLMs. This method ignores the different functionalities of attention heads, leading to the eviction of critical tokens and thus degrades the performance of LLMs.

To address the issue above, instead of using all the attention heads in GQA-based LLMs to determine important tokens as in the previous work, we first identify the attention heads in each layer that are not only capable of retrieving the initial and final tokens of a prompt, but also capable of retrieving important tokens within the text and attending to their surrounding semantic context. Afterwards, we exploit such heads to determine the important tokens and retain their corresponding KV cache pairs. Furthermore, we analyze the cache eviction error of each layer individually and introduce a layer-adaptive KV cache allocation strategy. Experimental results demonstrate the proposed CompressKV consistently outperforms state-of-the-art approaches under various memory budgets on LongBench and Needle-in-a-Haystack benchmarks. Notably, it retains over 97% of full-cache performance using only 3% of KV cache on LongBench's question-answering tasks and achieves 90% of accuracy with just 0.7% of KV storage on Needle-in-a-Haystack benchmark. Our code is available in the supplementary material.

## 1 Introduction

Recent advances in large language models (LLMs) (Achiam et al., 2024; Anthropic, 2024; Dubey et al., 2024; Hui et al., 2025; Wang et al., 2025) have boosted their long-context processing capabilities. However, with the increasing length of texts, the resulting key-value (KV) cache size grows linearly. The large KV cache leads to slow inference due to the attention calculation across past KV cache. In addition, the large KV cache requires substantial memory storage, which creates a major bottleneck in the deployment of long-context LLMs. Therefore, effective compression of KV cache is essential for optimizing the computational efficiency and model scalability.

State-of-the-art KV cache compression focuses on quantization, low-rank approximation, and KV cache eviction (Liu et al., 2024; Kang et al., 2024; Ge et al., 2024; Xiao et al., 2024; Li et al., 2024; Cai et al., 2025; Yang et al., 2024; Qin et al., 2025). Among such techniques, KV cache eviction strategy where KV pairs corresponding to those unimportant tokens are eliminated and the remaining KV pairs are kept has started to draw more and more attention.

There are different criteria to determine unimportant tokens for KV cache compression. For example, StreamingLLM (Xiao et al., 2024) retain the first and last tokens and neglects potentially important tokens in the middle of the prompt. SnapKV (Li et al., 2024) clusters recent attention scores within an observation window at the end of the prompt, either per head or per head group, to identify and retain the important tokens receiving the highest attention values. CAKE (Qin et al., 2025) extends SnapKV's method by adding the attention variance in an observation window to the eviction score, enabling it to capture tokens whose importance fluctuates over time.

The criteria described above are effective in many scenarios in KV cache compression. However, they treat all heads equally without examining their distinct functionalities, so that they use the sum of the attention scores across all the attention heads to make decisions on KV cache eviction. In fact, attention heads exhibit different functionalities. For example, in Grouped Query Attention (GQA)-based LLMs (Ainslie et al., 2023), some attention heads, called Streaming Heads, exclusively focus on the beginning and the end of a prompt (Xiao et al., 2024; 2025). When the attention heads within a GQA group are dominated by Streaming Heads, those heads have the largest influence on KV cache eviction, resulting in only the initial and last tokens' KV pairs being retained. This leads to the eviction of crucial tokens in the middle of a prompt and thus degrades performance of LLMs.

Besides eliminating KV pairs for those unimportant tokens, state-of-the-art research also allocates specified memory budgets to layers. For example, Xiao et al. (2024); Li et al. (2024) allocates each layer to a fixed number of KV pairs without considering layer difference. Yang et al. (2024); Cai et al. (2025); Qin et al. (2025) allocates KV cache budget across layers based on attention distributions or layer-wise statistics such as attention entropy or variance, which often require additional online computation cost. Moreover, since attention distributions can vary significantly across different models, limiting their generalization ability and effectiveness. Orthogonally, HeadKV (Fu et al., 2025) and AdaKV (Feng et al., 2025) extend to head-level budget allocation.

In this paper, we observe that certain attention heads are capable of retrieving important tokens within the text and attending to their surrounding semantic context. We refer to these heads as Semantic Retrieval Heads. Motivated by this observation, we identify such Semantic Retrieval Heads in each layer and use them to determine the crucial tokens and share a unified set of crucial token indices across all heads within that layer. This approach can substantially address the dominance of Streaming Heads in KV cache evictions, so that it can enhance the performance of GQA-based models. Furthermore, we analyze the cache eviction error of each layer individually and introduce a layer-adaptive KV cache allocation strategy. Our contributions are as follows:

(1) We introduce a Semantic-Retrieval–driven mechanism to address streaming-head dominance in GQA, preventing important tokens from being evicted out; The identified Semantic Retrieval Heads then guide token importance and KV-cache eviction. Our experimental results demonstrate Semantic Retrieval Heads know what tokens are unimportant before generation.

(2) We estimate each layer's compression impact by computing the Frobenius norm of the difference between its attention-block outputs with the compressed cache and those with the full cache, during the decoding stage. Cache budgets are then proportionally assigned across layers, prioritizing layers with higher errors. Importantly, this analysis is performed offline and does not introduce any additional overhead during online inference.

(3) CompressKV is validated on multiple LLMs using LongBench and Needle-in-a-Haystack (NIAH). On LongBench, CompressKV maintains over 99% of full-cache performance with only 19% of KV budget and retains 97% of question-answering accuracy using just 3% of the cache. On Needle-in-a-Haystack retrieval benchmark, it achieves 90% of the baseline accuracy with only 0.7% of KV storage.

## 2 BACKGROUND AND RELATED WORK

### 2.1 KV-CACHE BASICS

The motivation of KV cache is to reduce the signification computation cost of attention evaluation. To explain this, consider the case of a single attention head. This attention head can be evaluated with weight matrices, denoted as $\mathbf{W_Q}, \mathbf{W_K}, \mathbf{W_V} \in \mathbb{R}^{d \times d}$, and a prompt, denoted as $\mathbf{X} \in \mathbb{R}^{l \times d}$, where where $l$ is the sequence length and $d$ the hidden dimension. The attention evaluation includes two phases, i.e., prefilling phase and decoding phase.

*Prefilling Phase*: In this phase, the query $\mathbf{Q}$, key $\mathbf{K}$, and value $\mathbf{V}$ are evaluated with the entire input embeddings as follows

$$\mathbf{Q} = \mathbf{XW_Q}, \mathbf{K} = \mathbf{XW_K}, \mathbf{V} = \mathbf{XW_V} \tag{1}$$

With $\mathbf{K}$, $\mathbf{V}$ and $\mathbf{Q}$, the output of the attention can be evaluated as follows

$$\mathbf{O} = \mathrm{Softmax}(\mathbf{Q}\,\mathbf{K}^\top)\,\mathbf{V} \tag{2}$$

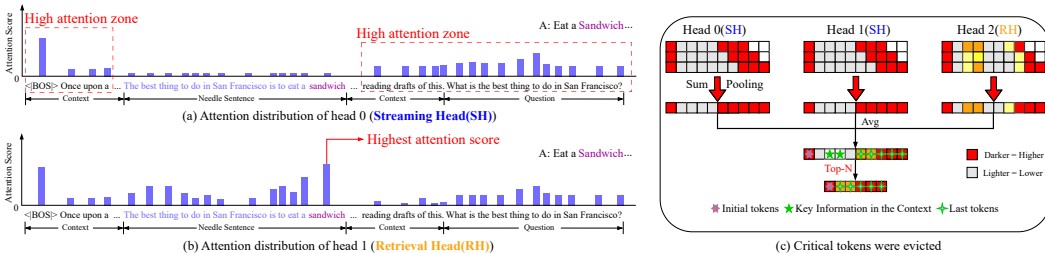

Figure 1: Motivation. (a) The attention score distribution of a streaming head (SH). (b) The attention score distribution of a retrieval head (RH). (c) Streaming attention heads in a GQA group dominate the token eviction, indicating only initial and final tokens are remained. The critical tokens are evicted.

The key $\mathbf{K}$ and the value $\mathbf{V}$ are then stored in cache memory, which is also called KV cache.

*Decoding Phase*: In this phase, the previously stored KV cache is used to generate new tokens and the newly generated KV pair is then appended to the previously stored KV cache to refresh KV cache. Specifically, at a decoding step $t$, given a new token embedding $x_t \in \mathbb{R}^{1 \times d}$, we first evaluate the newly generated KV pairs with this new token as follows

$$\mathbf{k_t} = x_t \, \mathbf{W_K}, \quad \mathbf{v_t} = x_t \, \mathbf{W_V}. \tag{3}$$

Afterwards, we use such new KV pairs to update the cache via

$$\mathbf{K} \leftarrow Concat\big[\, \mathbf{K}, \ \mathbf{k_t}\,\big], \mathbf{V} \leftarrow Concat\big[\, \mathbf{V}, \ \mathbf{v_t}\,\big]. \tag{4}$$

In GQA-based LLMs, query heads in a layer are partitioned into multiple groups. Multiple query heads within the same group share the same KV cache. The shared key and value are evaluated once per group and reused to produce the output of each head in the group. Although KV caching removes the need to recompute keys and values at every step, the cache itself grows linearly with prompt sequence length, becoming especially problematic for long-text tasks.

## 2.2 KV Cache Compression

To alleviate the burden of KV cache storage, various KV cache compression methods, e.g., quantization (Liu et al., 2024), low-rank approximations (Kang et al., 2024), and KV cache eviction strategy have been proposed. In particular, KV cache eviction reduces cache size by removing KV cache pairs of unimportant tokens without retraining. There are different eviction strategies. For example, StreamingLLM (Xiao et al., 2024) focuses solely on retaining the first and last tokens, which only addresses the Streaming Head scenario and neglects potentially important tokens in the middle of the sequence. To overcome this limitation, more advanced methods have been proposed (Liu et al., 2023; Zhang et al., 2023; Li et al., 2024; Han et al., 2024; Oren et al., 2024). A representative example is SnapKV (Li et al., 2024), which clusters recent attention scores, either per head or per head group to identify important token and retain the KV cache pairs of such tokens. Besides, recent approaches, including PyramidKV (Cai et al., 2025), D2O (Wan et al., 2025), and CAKE (Qin et al., 2025), dynamically allocate cache budgets based on attention statistics or modeled attention dynamics of all the layers in an LLM. Beyond layer-level allocation, HeadKV (Fu et al., 2025) and AdaKV (Feng et al., 2025) further enhances cache budget with head-level budget allocation. Their selection strategies for important tokens are an extended version of SnapKV's eviction strategy.

The KV cache eviction approaches above have two major limitations. First, In GQA-based LLMs, many prior KV cache eviction pipelines compute token importance via head-agnostic pooling (e.g., across heads within each GQA group) when selecting tokens for eviction, effectively treating all attention heads equally and ignoring their functional heterogeneity; Recent work (Olsson et al., 2022; Kwon et al., 2022; Zheng et al., 2024; Ren et al., 2024; Wu et al., 2025; Todd et al., 2024; Yin & Steinhardt, 2025; Tang et al., 2025; Fu et al., 2025) has shown that different attention heads have distinct roles. For example, some attention heads, called Streaming Heads in the state-of-the-art research, always focus on the beginning and the end of a prompt. For example, in Figure 1(a), head 0 is such a Streaming Head since the attention scores of the initial token and the last tokens are larger

than the remaining tokens. On the contrary, some attention heads, called Retrieval heads in Wu et al. (2025), exhibit copy-and-paste behaviors for long-context scenarios. For example, in Figure 1(b), head 1 is such a retrieval head since the attention scores of the correct answer "sandwich" are larger. HeadKV (Fu et al., 2025) further scores heads using retrieval and reasoning signals. In GQA-based LLMs, Streaming Heads tend to have larger effect than the other heads for KV cache eviction, which indicates only KV cache pairs corresponding to initial and last tokens are retained. This leads to the eviction of crucial tokens in the middle of a prompt and thus degrades the performance of LLMs. Figure 1(c) illustrates such an example, where Streaming Heads including head0 and head1 dominate token eviction for KV cache compression.

Second, the layer budget allocation in the previous work Yang et al. (2024); Cai et al. (2025); Qin et al. (2025) typically relies on attention distributions or layer-wise statistics such as attention entropy or variance, which often require additional online computation. Moreover, since attention distributions can vary significantly across different models, directly adopting a fixed allocation strategy according to attention distributions may not yield optimal results.

## 3 COMPRESSKV

CompressKV includes three key components: (1) Identification of the attention heads that are capable of retrieving important tokens within the text and attending to their surrounding semantic context. (2) Important token selection driven by such identified heads. (3) Error-aware layer-adaptive cache allocation. In the following subsections, we will first explain our observations and insights into identification of attention heads with specified functionalities. Afterwards, we will take advantage of such heads to select tokens for KV cache eviction. Furthermore, different cache budgets will be allocated to different layers.

### 3.1 OBSERVATIONS AND INSIGHTS

To avoid that Streaming Attention Heads dominate the KV cache eviction as illustrated in Figure 1(c), intuitively, Retrieval Heads instead of all attention heads can be used to identify important tokens for KV cache eviction. Previous work typically identifies Retrieval Heads using a strict top-1 rule, indicating that those attention heads, the highest attention score of which aligns exactly with the correct token answer during generation, are labeled as Retrieval Heads (Wu et al., 2025). This identification technique emphasizes copy-and-paste behavior. Tang et al. (2025) extends copy-and-paste identification by classifying both echo heads (copy-and-paste to the identical prior token) and induction heads (an extension that attends to the immediately preceding token) as Retrieval Heads. HeadKV (Fu et al., 2025) relaxes the strict top-1 criterion to a top-N hit: at each decoding step, a head is credited if the ground-truth answer token ranks within its top-k attention weights.

Although HeadKV are more relaxed than strict top-1, this criteria still remains peak-driven, privileging sharp attentions on the answer token. In long contexts where attention is sparse and skewed towards boundary tokens—top-1 rules yield low hit rates and can under-credit attention heads whose attention covers the answer span and its semantic neighborhood without placing a single sharp peak on the exact answer token. In HeadKV, if parts of the answer span do not appear within the top-k ranked positions, heads allocating substantial attention to these tokens may not be credited. For instance, in Figure 2(a), head 0 fails to receive credit because the relevant tokens fall outside the top-k range despite providing coverage around the correct answer. Moreover, because the top-k threshold in HeadKV is tied to the answer length, when answers are short, e.g., only one or two tokens, this method returns back to the original strict top-1 regime.

To address the limitation above, we introduce Semantic Retrieval Heads (SRH), a span-aggregation standard that credits attention heads for both copy-and-paste behaviours and deeper semantic dependencies. We then use such heads to identify important tokens for KV cache eviction, thereby preventing crucial mid-prompt evidence from being suppressed by streaming heads.

### 3.2 SEMANTIC RETRIEVAL HEAD IDENTIFICATION STANDARDS

Instead of requiring exact top-k hits in the traditional Retrieval Head identification, we aggregate a head's attention scores over the entire answer span inserted into a long context whenever the model

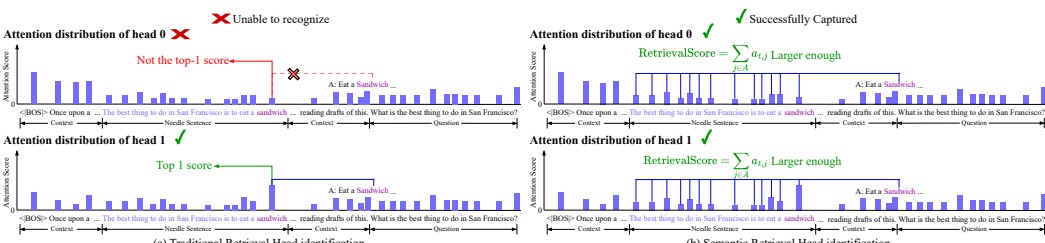

Figure 2: Illustration of Semantic Retrieval Head identification versus traditional Retrieval Head selection. Semantic Retrieval Heads capture attention over the entire answer span, addressing the limitations of traditional methods that rely solely on copy-and-paste behavior.

generates a correct answer token as the score of this head. This evaluation is expressed with the following equation as follows

$$\text{SemanticRetrievalScore}(h) = \sum_{t=1}^{N} \mathbf{1}_{\{y_t \in \mathcal{A}\}} \sum_{j \in \mathcal{A}} a_{t,j}^{(h)}. \tag{5}$$

where $y_t$ is the generated token at step $t$, $\mathcal{A}$ is the answer span, and $a_{t,j}^h$ is head $h$'s attention weight on the $j$-th token of $\mathcal{A}$. The higher the score of a head is, the more capable of capturing semantic information this head is.

Figure 2(b) illustrates the concept of this new identification standard. By summing over the entire span, we can capture attention heads that contribute semantically relevant context even when they never achieve top-1 attention on a single token. Aggregation over multiple tokens enables the method to recognize heads that attend to semantic cues—such as "eat" or "a thing" around "sandwich"—rather than only pure copy-and-paste patterns. For example, head 0 in Figure 2 is considered as Semantic Retrieval Head in our new standard although it is not considered as Retrieval Head in the traditional identification methods. For a visual comparison between Semantic Retrieval Heads and traditional Retrieval Heads, please refer to Appendix G

### 3.3 Token Selection Driven by Semantic Retrieval Heads

In GQA-based LLMs, for each layer, we will select top top-$k$ Semantic Retrieval Heads with high scores defined with equation (5) as the criterion for selecting important tokens for KV cache eviction. All the attention heads within this layer share a common set of selected token indices determined by these top Semantic Retrieval Heads. This concept is illustrated in Figure 3, where a layer has two groups. In this example, Head 2 and Head 3 are top 2 Semantic Retrieval Heads. The attention score matrices of such heads are compressed by summing over the observation window and pooling across the token dimension. Afterwards, such compressed vectors are averaged. The tokens with the top $N$ highest attention scores will be selected and their corresponding KV cache pairs will be retained. The KV cache pairs for the remaining tokens will be evicted to compress KV cache.

### 3.4 Error-Aware Layer-Adaptive Cache Allocation

To maximize memory efficiency under strict budget constraints, we propose an error-aware and layer-adaptive cache allocation strategy. Instead of relying on attention statistics as in the previous methods, this approach quantifies the compression error caused by KV cache compression, using full-cache outputs as the reference. We specifically focus on the extreme compression setting, where only a small fraction of tokens are retained in each layer's KV cache. For each layer $l$ and decoding step $t$, let $\mathbf{O}_{\text{full},t}^l$ and $\mathbf{O}_{\text{comp},t}^l$ denote the attention outputs using the full and compressed KV caches, respectively:

$$\mathbf{O}_{\text{full},t}^l = \mathbf{W}_O^l \, \text{Attention}\left(\mathbf{Q}_t^l, \, \mathbf{K}_{\text{full}}^l, \, \mathbf{V}_{\text{full}}^l\right) \tag{6}$$

$$\mathbf{O}_{\text{comp},t}^l = \mathbf{W}_O^l \, \text{Attention}\left(\mathbf{Q}_t^l, \, \mathbf{K}_{\text{comp}}^l, \, \mathbf{V}_{\text{comp}}^l\right) \tag{7}$$

where $\mathbf{W}_O^{(l)}$ is the output projection matrix of layer $l$, $\mathbf{Q}_t^l$ is the query, $\mathbf{K}^l$ is the key, and $\mathbf{V}^l$ is the value representation at layer $l$. To evaluate the error incurred by compressing KV cache per layer,

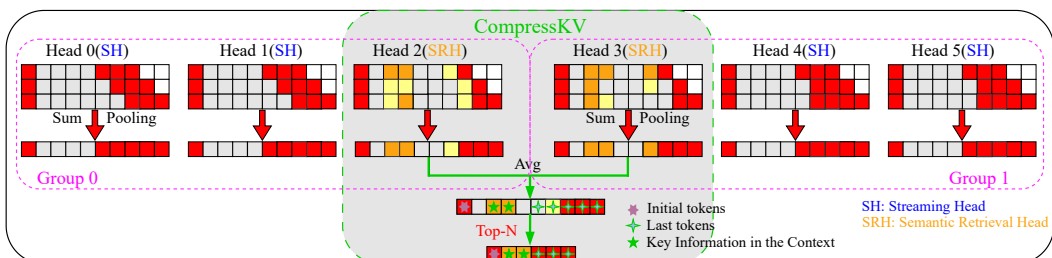

Figure 3: Illustration of the token selection driven by Semantic Retrieval Heads.

the error score for layer $l$ is computed and normalized as:

$$e^{(l)} = \sum_{t=1}^{T} \frac{\left\|\mathbf{O}_{\text{comp},t}^l - \mathbf{O}_{\text{full},t}^l\right\|_F}{\left\|\mathbf{O}_{\text{full},t}^l\right\|_F + \epsilon}, \tilde{e}^{(l)} = \frac{e^{(l)}}{\sum_k e^{(k)}} \tag{8}$$

where $T$ is the total number of decoding steps,$|\cdot|_F$ denotes the Frobenius norm and $\epsilon$ is a small positive constant (e.g., $10^{-6}$) to prevent division by zero.

Given the normalized per-layer error scores $\tilde{e}$ and total cache budget $B_{total}$, we first assign a minimum allocation $m$ and a maximum allocation $M$ to each layer to avoid a layer either has no memory budget or a large memory budget. The remaining budget is distributed in proportion to the error scores. More details can be found in Appendix C.

## 4 EXPERIMENTS

**Baselines and Backbone LLMs** We compare CompressKV with six representative work: StreamingLLM (Xiao et al., 2024), SnapKV (Li et al., 2024), PyramidKV (Cai et al., 2025), CAKE (Qin et al., 2025)),HeadKV (Fu et al., 2025) and AdaKV (Feng et al., 2025). All methods are evaluated on state-of-the-art open-source LLMs, including Llama-3.1-8B-Instruct (Dubey et al., 2024),Mistral-7B-Instruct-v0.3 (Jiang et al., 2024a), and Qwen2.5-7B-Instruct (Hui et al., 2025). In addition, we extend our evaluation to larger-scale LLMs, with detailed results provided in the Table 10. All evaluations are conducted in a generative setting using greedy decoding to ensure a fair comparison across tasks. Beyond direct comparison, we further demonstrate two orthogonal integrations: (i) CompressKV with head-level budget allocation methods in Appendix D and (ii) CompressKV with prefilling-stage acceleration methods in Appendix E

**Evaluating Tasks** To evaluate CompressKV's performance under different memory budgets, we adopt two comprehensive benchmarks and one masking-based ablation analysis: (1) Long-Bench (Bai et al., 2024), which evaluates long-context understanding across 16 datasets; see Appendix B for more details. (2) Needle-in-a-Haystack (Kamradt, 2023), which measures the retrieval of a target answer hidden in extended text; and (3) an ablation of retrieval head types (following Wu et al., 2024), where we selectively disable SRH and TRH to quantify their contributions. We also compare CompressKV with TRH vs. SRH under equal per-layer KV budgets, e.g., 256 tokens and report results separately.

**Implementation Details** Our experiments evaluate CompressKV and baseline methods under total memory budgets ranging from 128 to 2048 tokens for each layer. The KV cache budget is distributed equally across layers for baseline methods: StreamingLLM and SnapKV, while methods such as PyramidKV, CAKE, and CompressKV distributes the cache differently across layers but keeps total memory usage fixed. By contrast, HeadKV and AdaKV are head-level allocation schemes; in our setup, to respect grouped-query attention, we allocate at the GQA-group granularity under the same total memory. To ensure a fair comparison, tokens are evicted only during the prefilling phase. For CompressKV, we select the top four Semantic Retrieval Heads in each layer to identify and preserve the most important tokens. Using the LongBench benchmark, we derive each layer's normalized error scores by simulating minimal-size KV compression and computing the Frobenius-norm reconstruction error of its attention-block outputs. During budget allocation, we impose per-layer bounds $[m, M]$ with $m = 32$ and $M = 3 \times B_{\text{per-layer}}$, and distribute the remaining KV pairs proportionally to normalized errors.

Table 1: Performance comparison of CompressKV with StreamingLLM, SnapKV, PyramidKV, CAKE, HeadKV, AdaKV, and FullKV on LongBench for Llama-3.1-8B-Instruct, Mistral-7B-Instruct-v0.3 and Qwen2.5-7B-Instruct. CompressKV generally outperforms other KV cache compression methods across various KV cache sizes and LLMs. Best in **bold**

| Method | Single-Document QA | | | Multi-Document QA | | | Summarization | | | Few-shot Learning | | | Synthetic | | Code | | Avg. |
|---|---|---|---|---|---|---|---|---|---|---|---|---|---|---|---|---|---|
| | NrtvQA | Qasper | MF-en | HotpotQA | Musique | 2WikiMQA | GovReport | QMSum | MultiNews | TREC | TriviaQA | SAMSum | PCount | PR-en | Lcc | RB-P | |
| Llama3.1-8B-Instruct, KV Size = 256 | | | | | | | | | | | | | | | | | |
| StreamingLLM | 23.35 | 22.26 | 33.94 | 42.8 | 21.91 | 24.47 | 23.35 | 16.28 | 23.85 | 54.5 | 70.66 | 17.65 | 6.12 | 87.99 | 39.64 | 34.01 | 33.92 |
| SnapKV | 30.41 | 36.24 | 49.88 | 54.06 | 30.69 | 45.95 | 23.64 | 22.99 | 23.6 | 58.0 | 91.29 | 40.92 | 6.25 | 99.0 | 59.68 | 50.74 | 45.21 |
| PyramidKV | 29.02 | 33.4 | 49.43 | 54.79 | 29.21 | 46.24 | 23.45 | 22.87 | 22.81 | 58.0 | 89.41 | 39.8 | 6.25 | 98.5 | 57.42 | 49.16 | 44.36 |
| CAKE | 31.31 | 40.18 | 51.54 | 54.48 | 29.66 | 45.75 | 25.22 | 23.84 | 24.09 | 64.5 | 91.49 | 42.06 | 6.14 | 99.5 | 59.6 | 51.51 | 46.3 |
| HeadKV(GQA) | 30.55 | 32.41 | 48.36 | 53.72 | 30.26 | 45.97 | 23.93 | 23.57 | 23.4 | 57.0 | 90.0 | 40.54 | 6.33 | 93.0 | 58.23 | 48.45 | 44.11 |
| AdaKV(GQA) | 29.45 | 33.7 | 48.96 | 54.48 | 29.59 | 46.09 | 23.58 | 23.13 | 23.27 | 53.0 | 90.79 | 40.49 | 6.33 | 98.5 | 59.47 | 50.31 | 44.45 |
| CompressKV | 30.68 | 42.58 | 52.27 | 54.37 | 30.43 | 46.46 | 24.86 | 23.79 | 24.13 | 68.5 | 89.97 | 41.08 | 6.14 | 99.5 | 60.63 | 51.94 | **46.71** |
| Llama3.1-8B-Instruct, KV Size = 1024 | | | | | | | | | | | | | | | | | |
| StreamingLLM | 23.55 | 28.99 | 43.17 | 42.5 | 21.61 | 28.37 | 28.26 | 19.12 | 26.37 | 68.0 | 74.02 | 19.41 | 6.19 | 82.58 | 43.65 | 35.49 | 36.95 |
| SnapKV | 31.32 | 44.62 | 52.51 | 54.65 | 30.24 | 46.8 | 28.14 | 24.12 | 26.36 | 69.0 | 92.05 | 42.67 | 6.12 | 99.5 | 62.06 | 54.99 | 47.82 |
| PyramidKV | 31.06 | 44.09 | 53.25 | 54.25 | 30.45 | 46.87 | 27.44 | 23.96 | 26.31 | 70.0 | 91.93 | 42.91 | 6.08 | 99.5 | 62.03 | 52.75 | 47.65 |
| CAKE | 30.61 | 44.64 | 52.18 | 54.89 | 30.44 | 46.14 | 29.1 | 24.28 | 26.33 | 71.0 | 91.89 | 42.81 | 6.17 | 99.5 | 62.05 | 55.47 | 47.97 |
| HeadKV(GQA) | 30.6 | 41.24 | 52.13 | 54.34 | 30.96 | 46.33 | 27.27 | 24.11 | 26.0 | 65.5 | 91.51 | 43.1 | 6.17 | 98.5 | 61.84 | 53.17 | 47.05 |
| AdaKV(GQA) | 31.75 | 44.71 | 52.74 | 54.75 | 31.05 | 46.73 | 27.9 | 23.86 | 26.41 | 69.5 | 92.05 | 42.82 | 6.58 | 99.5 | 62.04 | 54.61 | 47.94 |
| CompressKV | 30.27 | 45.18 | 53.44 | 55.25 | 30.83 | 46.57 | 29.06 | 24.34 | 26.48 | 71.0 | 91.79 | 43.37 | 6.0 | 99.5 | 63.14 | 55.62 | **48.24** |
| Mistral-7B-Instruct-v0.3, KV Size = 256 | | | | | | | | | | | | | | | | | |
| StreamingLLM | 21.89 | 21.37 | 32.51 | 36.98 | 16.78 | 25.43 | 21.85 | 16.89 | 23.55 | 57.5 | 72.38 | 18.24 | 4.0 | 65.0 | 31.63 | 33.53 | 31.22 |
| SnapKV | 27.49 | 29.18 | 48.92 | 48.0 | 26.47 | 36.78 | 22.46 | 22.04 | 22.54 | 70.0 | 89.44 | 43.72 | 6.0 | 96.0 | 56.69 | 54.49 | 43.76 |
| PyramidKV | 27.71 | 28.23 | 48.25 | 48.61 | 25.62 | 36.18 | 22.1 | 21.77 | 21.8 | 70.0 | 89.33 | 43.65 | 6.0 | 93.5 | 55.13 | 51.06 | 43.06 |
| CAKE | 27.46 | 33.09 | 54.32 | 48.32 | 27.08 | 37.8 | 24.62 | 23.61 | 23.85 | 70.0 | 89.33 | 44.09 | 4.5 | 95.5 | 55.76 | 56.35 | 44.73 |
| HeadKV(GQA) | 29.15 | 31.66 | 51.03 | 49.09 | 24.88 | 36.64 | 22.46 | 22.2 | 22.97 | 70.5 | 89.57 | 43.86 | 5.5 | 94.0 | 56.77 | 55.39 | 44.1 |
| AdaKV(GQA) | 29.08 | 27.93 | 49.69 | 49.4 | 26.07 | 36.34 | 22.55 | 21.72 | 22.14 | 70.5 | 89.69 | 44.09 | 6.5 | 94.0 | 56.21 | 54.05 | 43.75 |
| CompressKV | 29.86 | 35.32 | 52.84 | 49.77 | 27.64 | 38.04 | 24.05 | 23.37 | 23.26 | 75.5 | 89.16 | 45.31 | 4.5 | 96.5 | 56.47 | 55.31 | **45.43** |
| Mistral-7B-Instruct-v0.3, KV Size = 1024 | | | | | | | | | | | | | | | | | |
| StreamingLLM | 22.37 | 28.03 | 41.21 | 38.0 | 17.05 | 26.95 | 27.75 | 19.87 | 27.13 | 71.5 | 70.34 | 19.02 | 5.37 | 68.5 | 37.95 | 34.57 | 34.73 |
| SnapKV | 29.82 | 36.49 | 52.64 | 50.33 | 26.88 | 38.53 | 26.39 | 23.94 | 25.44 | 75.0 | 89.24 | 46.73 | 6.5 | 97.0 | 58.57 | 59.86 | 46.48 |
| PyramidKV | 28.14 | 35.83 | 54.28 | 49.88 | 25.68 | 38.31 | 25.91 | 23.82 | 25.42 | 74.5 | 89.86 | 46.17 | 5.0 | 97.5 | 57.72 | 57.36 | 45.96 |
| CAKE | 29.21 | 36.86 | 53.2 | 48.69 | 27.62 | 38.78 | 28.94 | 24.54 | 26.98 | 74.0 | 88.94 | 46.94 | 5.0 | 98.0 | 58.88 | 59.91 | 46.66 |
| HeadKV(GQA) | 29.62 | 37.34 | 53.09 | 50.31 | 26.67 | 38.28 | 26.44 | 23.52 | 26.11 | 76.0 | 89.35 | 45.68 | 5.0 | 97.0 | 58.54 | 59.56 | 46.41 |
| AdaKV(GQA) | 29.73 | 37.04 | 53.07 | 50.01 | 26.32 | 37.54 | 26.27 | 23.97 | 26.12 | 75.5 | 89.49 | 46.25 | 6.5 | 96.5 | 58.62 | 59.16 | 46.38 |
| CompressKV | 29.75 | 38.88 | 52.81 | 49.71 | 28.48 | 39.04 | 28.26 | 24.95 | 26.88 | 76.0 | 89.24 | 46.16 | 5.5 | 97.0 | 58.65 | 60.05 | **46.96** |
| Qwen2.5-7B-Instruct, KV Size = 256 | | | | | | | | | | | | | | | | | |
| StreamingLLM | 18.04 | 21.45 | 27.37 | 35.88 | 14.17 | 19.45 | 22.23 | 15.54 | 20.77 | 52.5 | 62.65 | 17.86 | 8.5 | 25.33 | 33.76 | 33.51 | 26.81 |
| SnapKV | 27.32 | 35.61 | 49.31 | 54.66 | 27.11 | 43.67 | 21.93 | 20.88 | 19.33 | 53.0 | 88.04 | 43.2 | 8.5 | 98.0 | 55.08 | 56.46 | 43.88 |
| PyramidKV | 25.05 | 33.19 | 47.72 | 52.62 | 25.28 | 44.05 | 20.14 | 19.92 | 16.64 | 50.5 | 87.76 | 40.71 | 8.5 | 94.0 | 50.36 | 51.77 | 41.76 |
| CAKE | 26.56 | 36.42 | 49.88 | 54.25 | 27.56 | 45.93 | 23.22 | 20.82 | 19.82 | 53.0 | 86.08 | 43.59 | 8.5 | 97.0 | 54.44 | 57.81 | 44.05 |
| HeadKV(GQA) | 27.62 | 36.57 | 49.71 | 54.96 | 26.82 | 44.83 | 22.51 | 20.43 | 19.5 | 56.5 | 89.12 | 43.43 | 8.5 | 98.5 | 55.63 | 58.33 | 44.56 |
| AdaKV(GQA) | 27.19 | 35.54 | 48.15 | 54.34 | 27.8 | 44.7 | 21.74 | 20.44 | 18.99 | 53.5 | 87.65 | 42.5 | 8.5 | 98.0 | 55.04 | 56.65 | 43.8 |
| CompressKV | 28.94 | 36.39 | 48.85 | 54.45 | 26.65 | 44.13 | 22.37 | 20.96 | 19.02 | 65.5 | 87.5 | 42.08 | 8.5 | 98.5 | 53.72 | 57.46 | **44.69** |
| Qwen2.5-7B-Instruct, KV Size = 1024 | | | | | | | | | | | | | | | | | |
| StreamingLLM | 20.68 | 30.34 | 41.03 | 37.96 | 16.35 | 25.71 | 27.04 | 17.89 | 23.39 | 67.0 | 61.54 | 18.18 | 8.5 | 12.5 | 39.15 | 36.4 | 30.23 |
| SnapKV | 28.77 | 40.73 | 51.54 | 56.84 | 28.64 | 45.65 | 26.7 | 21.87 | 22.84 | 69.0 | 89.57 | 44.35 | 8.5 | 99.5 | 59.65 | 64.85 | 47.44 |
| PyramidKV | 29.71 | 39.75 | 52.22 | 56.39 | 26.09 | 45.77 | 24.57 | 20.91 | 21.19 | 68.5 | 89.14 | 44.06 | 8.5 | 99.0 | 58.44 | 60.03 | 46.52 |
| CAKE | 28.88 | 42.2 | 51.66 | 56.66 | 28.96 | 45.25 | 28.0 | 21.98 | 23.09 | 68.5 | 88.73 | 44.85 | 8.5 | 100.0 | 58.95 | 64.15 | 47.52 |
| HeadKV(GQA) | 28.6 | 41.32 | 52.07 | 57.09 | 28.98 | 45.32 | 27.39 | 21.91 | 23.11 | 69.5 | 89.58 | 44.58 | 8.5 | 99.5 | 59.56 | 64.47 | 47.59 |
| AdaKV(GQA) | 28.9 | 40.86 | 52.09 | 56.93 | 26.62 | 45.64 | 26.51 | 21.87 | 22.89 | 69.5 | 89.74 | 44.44 | 8.5 | 99.5 | 59.83 | 63.32 | 47.32 |
| CompressKV | 29.52 | 42.17 | 51.82 | 57.55 | 29.5 | 45.17 | 26.96 | 22.06 | 22.69 | 70.5 | 88.43 | 44.11 | 8.5 | 100.0 | 59.08 | 63.94 | **47.62** |

## 4.1 EVALUATION ON LONGBENCH BENCHMARK

Table 1 demonstrates performance comparison under two KV cache budgets—low (256) and high (1024)—with full results across additional budgets as well as large-scale language model evaluations reported in Appendix F. CompressKV consistently ranks the top performers across various tasks. The advantage of CompressKV is particularly pronounced in low-memory scenarios. CompressKV outperforms HeadKV by 2.5 points on Llama-3.1-8B-Instruct; even in the 1024 cache budget setting scenario, CompressKV still maintains superior accuracy. By leveraging a small number of Semantic Retrieval Heads to accurately identify semantically important tokens, combined with an effective adaptive layer budget allocation strategy, CompressKV achieves the best overall performance.

As illustrated in Figure 4, we benchmark CompressKV on LongBench across KV cache sizes from 128 to 2048, presenting results for Llama-3.1-8B-Instruct, Mistral-7B-Instruct-v0.3, and Qwen-2.5-7B-Instruct. The evaluation metric is the average score across all LongBench datasets. SnapKV outperforms the legacy method StreamingLLM. Despite its methodological similarities to SnapKV, PyramidKV underperforms in many scenarios, possibly due to its limited adaptability. CAKE achieves better results than previous baseline methods in most cases by dynamically allocating memory to each layer and incorporating additional computations of variance and entropy scores. HeadKV and AdaKV (head-level allocation) perform strongly at generous budgets but degrade under tight budgets. In contrast, CompressKV consistently surpasses all methods across budgets—with

the largest margins in the low-budget regime—except for Qwen-2.5-7B-Instruct at a per-layer budget of 2048 tokens, where it is on par with HeadKV; in all other model–budget combinations, CompressKV attains the best average score.

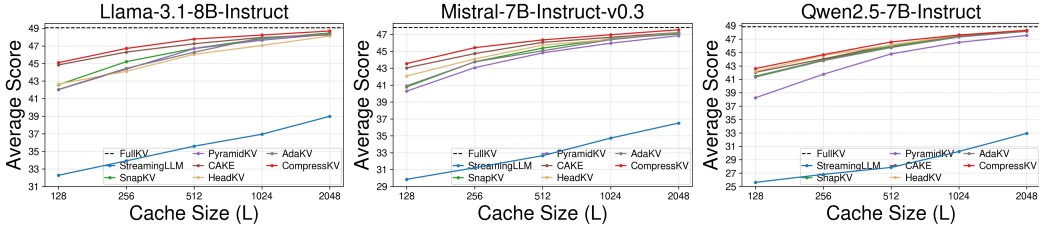

Figure 4: Average performance on 16 LongBench datasets under different KV cache budget settings compared with various baseline methods.

## 4.2 EVALUATION ON NEEDLE IN A HAYSTACK

Figure 5 presents average Needle-in-a-Haystack performance across KV budgets for three LLMs—8K–128K contexts for Llama-3.1-8B-Instruct and Qwen-2.5-7B-Instruct, and 2K–32K for Mistral-7B—showing CompressKV consistently surpasses competing methods at every budget. On Mistral-7B-Instruct-v0.3, CompressKV, HeadKV, and CAKE achieve near lossless compression with as few as 256 KV budget, highlighting their robustness. On Qwen-2.5-7B-Instruct, however, CAKE lags under low budgets, whereas CompressKV remains competitive. On Llama-3.1-8B-Instruct, AdaKV and HeadKV also underperform at low budgets, while CompressKV achieves nearly lossless performance at a 2048 KV budget (5% of the full cache) and still retains 90% of the original performance with only 256 KV budget (0.7% capacity). Together with the LongBench evaluation, these results show that CompressKV preserves general LLM performance across diverse long-context tasks while delivering efficient KV-cache compression. Additional results appear in Appendix H.

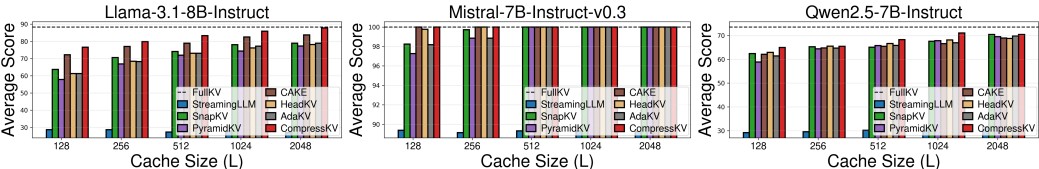

Figure 5: Average performance on the Needle-in-a-Haystack benchmark under different KV cache budget settings, in comparison with baseline methods.

## 4.3 SEMANTIC RETRIEVAL HEADS: CAUSAL ABLATION AND HEAD-AGNOSTIC GAINS

Following the masking-based causal test of Wu et al. (2025), we perform targeted ablation by masking the top 20 of these heads and comparing against traditional Retrieval Heads(TRH), as shown in Figure 6 . Even masking a small subset of Semantic Retrieval Heads causes a sharp drop in retrieval accuracy and a significant rise in hallucinations, underscoring their essential role in preserving factual consistency and their ability to retrieve and localize textual information. For more results, please refer to the Appendix I. CompressKV is compatible with heterogeneous head definitions. Table 2 compares CompressKV using TRH vs. SRH under a fixed per-layer KV budget of 256 tokens. SRH yields a modest yet consistent average gain over TRH (+0.24). Moreover, even with TRH and without dynamic budget allocation, CompressKV still surpasses most representative baselines (Table 1), evidencing more precise salient-token selection.

## 4.4 EVALUATION OF LATENCY AND PEAK MEMORY

We evaluate the end-to-end generation latency, decoding latency, and peak memory usage on Llama-3.1-8B-Instruct, implemented with FlashAttention-2 (Dao, 2024), running on a single NVIDIA A100 GPU. The evaluation spans context lengths from 4K to 128K tokens with a fixed generation length of 1024 tokens. We compare our proposed CompressKV method against a full cache

Table 2: LongBench accuracy under a fixed per-layer KV budget (256) comparing TRH vs. SRH.

| Method | NrtvQA | Qasper | MF-en | HotpotQA | Musique | 2WikiMQA | GovReport | QMSum | MultiNews | TREC | TriviaQA | SAMSum | PCount | PR-en | Lcc | RB-P | Avg |
|---|---|---|---|---|---|---|---|---|---|---|---|---|---|---|---|---|---|
| | | | | | | Mistral-7B-Instruct-v0.3, KV Size = 256 | | | | | | | | | | | |
| CompressKV(TRH) | 28.67 | 32.47 | 53.45 | 50.03 | 27.68 | 37.04 | 23.28 | 23.12 | 22.82 | 73.0 | 90.33 | 44.24 | 4.5 | 95.5 | 54.99 | 54.4 | 44.72 |
| CompressKV(SRH) | 28.99 | 34.6 | 54.65 | 49.17 | 27.58 | 37.94 | 23.24 | 23.23 | 22.71 | 74.5 | 89.28 | 44.07 | 3.5 | 95.5 | 55.83 | 54.51 | **44.96** |

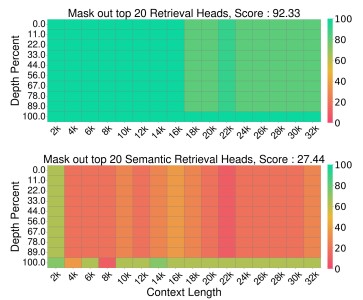

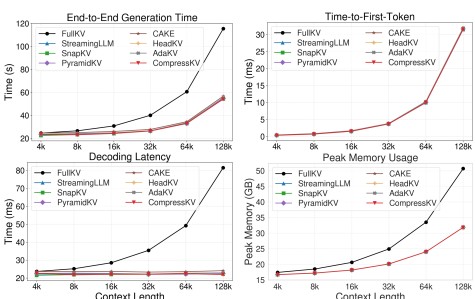

Figure 6: Masking different head types in Mistral-7B-Instruct-v0.3.

Figure 7: Comprehensive evaluation of inference efficiency on a single NVIDIA A100 GPU.

baseline and six aforementioned KV cache eviction methods—each constrained by a KV cache budget of 1024. As illustrated in Figure 7, the end-to-end generation latency and time-to-first-token increases with longer context lengths for all methods. However, while the decoding latency of the full-cache baseline consistently grows with context length, all KV cache eviction strategies—including CompressKV—maintain nearly constant decoding latency, demonstrating their efficiency. Figure 7 shows that with a fixed KV budget, all eviction methods (including CompressKV) have similar peak memory, while the full-cache baseline is much higher—especially at long contexts.

## 4.5 ABLATION STUDIES

To evaluate the effectiveness of each part in CompressKV, we conduct a series of ablation studies on the LongBench benchmark using Mistral-7B-Instruct-v0.3 with a fixed KV cache budget of 256.

**Ablation Study on the Number of Selected Heads per Layer** To quantify how many Semantic Retrieval Heads are needed per layer, we vary per-layer SR-Head selection from 2 to 24 on Table 3. Accuracy gains peak at 4 heads and then plateau (Top-6: $-0.17$; Top-12: $0.00$), with 24 heads slightly worse. Thus, 4 heads are sufficient to capture most semantic-retrieval capacity.

Table 3: Ablation studies: (Left) number of Semantic Retrieval Heads per layer; (Right) token selection strategy and layer-aware cache allocation.

| Heads per Layer | Mean Acc. (%) | Δ vs. Top-4 (%) |
|---|---|---|
| Top-2 | 44.33 | –0.63 |
| Top-4 | 44.96 | 0.00 |
| Top-6 | 44.79 | –0.17 |
| Top-12 | 44.96 | 0.00 |
| Top-24 | 44.30 | –0.66 |

| Method | Acc. (%) |
|---|---|
| SnapKV | 43.76 |
| + SRH Selection | 44.96 |
| + SRH + Layer Alloc | 45.43 |

**Ablation Study on Token Selection and Layer-Wise Cache Allocation** We ablate SR-Head–driven token selection and layer-aware budget allocation on Table 3). Adding our selection to SnapKV improves accuracy; adding layer-aware allocation yields further gains—both components are complementary.

## 5 CONCLUSION

In this work, we proposed CompressKV, a novel KV-cache compression framework for GQA-based LLMs that (1) addresses streaming-head dominance via Semantic Retrieval Heads guide token-importance estimation and KV eviction, evicting unimportant tokens before generation and (2) allocates a layer-adaptive cache budget by measuring each layer's offline cache-eviction error. Extensive experiments on LongBench and Needle-in-a-Haystack across multiple model architectures and cache budgets confirm better performance under diverse memory constraints.

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

## A    LLM Usage Statement

In accordance with the ICLR policy on the use of Large Language Models (LLMs), we disclose that ChatGPT 5 (by OpenAI) was used solely as a writing assistant to polish the language and improve the readability of the paper. It was not involved in research ideation, experimental design, implementation, data analysis, or result interpretation.

## B    Dataset Details

Table 4 presents the LongBench benchmark used in our experiments, which consists of 14 English subtasks and 2 code-completion subtasks organized into six categories—single-document QA, multi-document QA, summarization, few-shot learning, synthetic tasks, and code completion. Each subtask contains 150–500 samples with input lengths ranging from 1,235 to 18,409 words. Evaluation metrics include F1, Rouge-L, classification accuracy, and edit similarity.

Table 4: An overview of the dataset statistics in LongBench.

| Dataset | Source | Task Type | Avg Len | Metric | Language | # Samples |
|---|---|---|---|---|---|---|
| NarrativeQA | Literature, Film | Single-Document QA | 18,409 | F1 | English | 200 |
| Qasper | Science | Single-Document QA | 3,619 | F1 | English | 200 |
| MultiFieldQA-en | Multi-field | Single-Document QA | 4,559 | F1 | English | 150 |
| HotpotQA | Wikipedia | Multi-Document QA | 9,151 | F1 | English | 200 |
| 2WikiMultihopQA | Wikipedia | Multi-Document QA | 4,887 | F1 | English | 200 |
| MuSiQue | Wikipedia | Multi-Document QA | 11,214 | F1 | English | 200 |
| GovReport | Government report | Summarization | 8,734 | Rouge-L | English | 200 |
| QMSum | Meeting | Summarization | 10,614 | Rouge-L | English | 200 |
| MultiNews | News | Summarization | 2,113 | Rouge-L | English | 200 |
| TREC | Web question | Few-shot Learning | 5,177 | Accuracy (CLS) | English | 200 |
| TriviaQA | Wikipedia, Web | Few-shot Learning | 8,209 | F1 | English | 200 |
| SAMSum | Dialogue | Few-shot Learning | 6,258 | Rouge-L | English | 200 |
| PassageCount | Wikipedia | Synthetic Task | 11,141 | Accuracy (EM) | English | 200 |
| PassageRetrieval-en | Wikipedia | Synthetic Task | 9,289 | Accuracy (EM) | English | 200 |
| LCC | Github | Code Completion | 1,235 | Edit Sim | Python/C#/Java | 500 |
| RepoBench-P | Github repository | Code Completion | 4,206 | Edit Sim | Python/Java | 500 |

# C  MORE IMPLEMENTATION DETAILS

In this section, we provide additional details of our experimental setup and a comprehensive description of the error-aware, layer-adaptive cache allocation algorithm used by CompressKV. To ensure a fair comparison across all KV cache compression methods, we use identical hyperparameters: an observation window of 8 tokens, a 1D pooling kernel of size 5, and average-pooling to aggregate attention scores.

## C.1  DETAILED DESCRIPTION OF ERROR-AWARE LAYER-ADAPTIVE CACHE ALLOCATION

Using the LongBench benchmark, we simulate an extreme compression scenario by restricting each layer's KV cache size to 32 tokens (approximately 0.3% of full capacity). Unlike completely skipping an attention block (binary on/off), retaining a small subset of tokens allows us to explicitly quantify the direct impact of KV cache compression on the attention outputs. This approach effectively captures fine-grained compression errors without incurring multiple forward computations that would otherwise be necessary for evaluating the complete removal of attention blocks.

Formally, for each dataset $d \in D$, transformer layer $l$, and decoding step $t$, we compute the per-layer compression-induced reconstruction error as follows:

$$e_d^{(l)} = \sum_{t=1}^{T} \frac{\|\mathbf{O}_{\text{comp},t}^{(l)} - \mathbf{O}_{\text{full},t}^{(l)}\|_F}{\|\mathbf{O}_{\text{full},t}^{(l)}\|_F + \epsilon} \tag{9}$$

where $T$ denotes the total decoding steps, $\|\cdot\|_F$ represents the Frobenius norm, and $\epsilon = 10^{-6}$ ensures numerical stability. Next, we perform an L1 normalization of the per-layer errors within each dataset:

$$\hat{e}_d^{(l)} = \frac{e_d^{(l)}}{\sum_k e_d^{(k)}}. \tag{10}$$

Then, we average these normalized per-layer errors across all datasets:

$$\bar{e}^{(l)} = \frac{1}{|D|} \sum_{d \in D} \hat{e}_d^{(l)}. \tag{11}$$

Finally, we apply another L1-normalization across layers to obtain the final importance scores:

$$\tilde{e}^{(l)} = \frac{\bar{e}^{(l)}}{\sum_k \bar{e}^{(k)}}. \tag{12}$$

Averaging normalized errors across all datasets ensures both generalizability and fairness: by averaging errors from diverse datasets, we capture consistent trends in layer importance rather than

overfitting to any single task or domain. Compared with budget allocation methods that rely solely on attention-score distributions, our error-aware approach explicitly quantifies the impact of compression on the model's final attention outputs, resulting in a more precise and effective allocation strategy. These normalized, dataset-averaged error scores $\tilde{e}^{(l)}$ guide our error-aware, layer-adaptive cache allocation as detailed in Algorithm 1 below.

To safeguard against extreme cases, we impose per-layer bounds $[m, M]$, where the minimum allocation $m = 32$ ensures that each layer receives at least a small, baseline cache allocation, preventing any single layer from becoming completely inactive under extreme conditions. The upper bound $M = 3 \times B_{\text{per-layer}}$ prevents excessive cache allocation to any individual layer, ensuring a balanced distribution of cache resources and maintaining overall model performance. Additionally, we plot the performance of both the Mistral-7B-Instruct-v0.3 and Llama-3.1-8B-Instruct models under a per-layer KV cache budget of 256 tokens as bar charts (see Figures 8), illustrating the distinct allocation characteristics of each model.

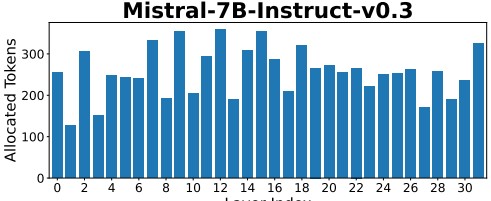
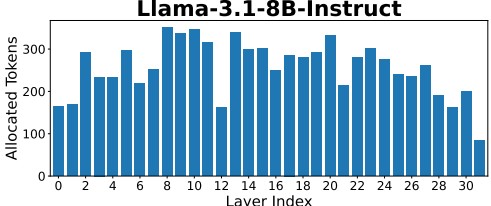

Figure 8: Per-layer KV cache allocation for Mistral-7B-Instruct-v0.3 (left) and Llama-3.1-8B-Instruct (right) under a total budget of 256 tokens per layer.

---

**Algorithm 1** Error-aware Layer-adaptive Cache Allocation

---

**Require:** Scores $\tilde{e}$, total budget $B_{\text{total}}$, per-layer bounds $[m, M]$
**Ensure:** Allocations $\mathbf{B}$
1: $B_i \leftarrow m, \forall i$
2: $R \leftarrow B_{\text{total}} - \sum_i B_i$
3: $B_i \leftarrow \text{clip}(B_i + \text{round}(\tilde{e}_i \cdot R), m, M), \forall i$
4: $\Delta \leftarrow B_{\text{total}} - \sum_i B_i$
5: **while** $\Delta \neq 0$ **do**
6:   **if** $\Delta > 0$ **then**
7:     $\mathcal{L} \leftarrow \{i \mid B_i < M\}$
8:     **if** $\mathcal{L} = \emptyset$ **then**
9:       Break
10:     **end if**
11:     $j \leftarrow \arg\max_{i \in \mathcal{L}} \tilde{e}_i, B_j \leftarrow B_j + 1, \Delta \leftarrow \Delta - 1$
12:   **else**
13:     $\mathcal{L} \leftarrow \{i \mid B_i > m\}$
14:     **if** $\mathcal{L} = \emptyset$ **then**
15:       Break
16:     **end if**
17:     $j \leftarrow \arg\min_{i \in \mathcal{L}} \tilde{e}_i, B_j \leftarrow B_j - 1, \Delta \leftarrow \Delta + 1$
18:   **end if**
19: **end while**
20: **return** $\mathbf{B}$

---

# D ORTHOGONAL INTEGRATION WITH HEAD-LEVEL BUDGET ALLOCATION METHODS

CompressKV is orthogonal to head-level budget allocation methods such as HeadKV (Jiang et al., 2024b) and AdaKV (Feng et al., 2025). To examine complementarity, we graft our components onto these backbones:

- HeadCompressKV = HeadKV + our token selection;
- AdaCompressKV = AdaKV + our token selection and error-aware, layer-wise budget allocation.

Evaluated on LongBench with Llama-3.1-8B-Instruct and Mistral-7B-Instruct-v0.3 (Table 5), both HeadCompressKV and AdaCompressKV outperform their respective backbones, with the largest gains under tight KV budgets. Consistent patterns hold on the Needle-in-a-Haystack benchmark (Figure 9). On Llama-3.1-8B-Instruct at low KV budgets (e.g., 128 cache budget per layer), Head-CompressKV improves over HeadKV by 6% relative accuracy, and AdaCompressKV improves over AdaKV by 13%.

These results indicate that combining CompressKV's token selection with error-aware budget allocation further strengthens head-level allocation schemes by avoiding GQA-induced overwriting during eviction and prioritizing capacity where it is most impactful—especially in the low-budget regime.

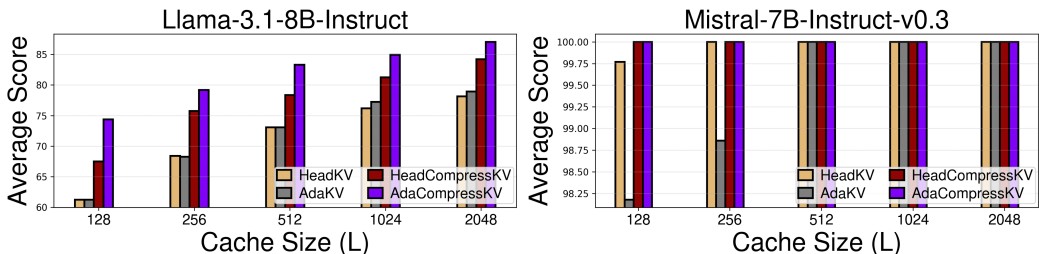

Figure 9: Needle-in-a-Haystack test results on Llama-3.1-8B-Instruct and Mistral-7B-Instruct-v0.3.

# E    ORTHOGONAL INTEGRATION WITH PREFILLING-STAGE ACCELERATION METHODS

CompressKV is orthogonal to prefilling-stage acceleration approaches such as MInference (Jiang et al., 2024b) and XAttention (Xu et al., 2025). While MInference and XAttention primarily speed up long-context inference during the prefilling stage via sparse attention, CompressKV targets the decoding stage by reducing the KV-cache footprint, thereby alleviating the memory-bound bottleneck and improving throughput.

To verify this complementarity, we integrate CompressKV with MInference and XAttention and evaluate on the LongBench benchmark; results are summarized in Table 6 under a KV budget of 2048 tokens per layer. For MInference, we adopt its default configuration; for XAttention, we use stride=8 and threshold=0.9. Across tasks, the combined variants maintain accuracy within a narrow band relative to their prefilling-only counterparts, while enabling decoding-stage memory reduction—collaborating that the two classes of techniques address orthogonal bottlenecks.

# F    COMPREHENSIVE RESULTS ON THE LONGBENCH DATASET

In Table 7- 9, we provide the detailed results corresponding to Figure 4 in the main paper. Across all KV cache budgets, CompressKV consistently outperforms the baseline methods, with one exception: on Qwen-2.5-7B-Instruct at a per-layer budget of 2048 tokens, it performs on par with HeadKV. The performance advantage of CompressKV becomes especially pronounced under tight memory constraints (i.e., smaller cache sizes). We further extend our evaluation to large-scale LLMs, including Qwen2.5-14B-Instruct and Qwen2.5-32B-Instruct, and report the results in Table 10, which show consistent improvements.

Table 5: Detailed results of integrating CompressKV with head-level allocation methods on Long-Bench with Llama-3.1-8B-Instruct and Mistral-7B-Instruct-v0.3.

| Method | Single-Document QA | | | Multi-Document QA | | | Summarization | | | Few-shot Learning | | | Synthetic | | Code | | Avg. |
|---|---|---|---|---|---|---|---|---|---|---|---|---|---|---|---|---|---|
| | NrtvQA | Qasper | MF-en | HotpotQA | Musique | 2WikiMQA | GovReport | QMSum | MultiNews | TREC | TriviaQA | SAMSum | PCount | PR-en | Lcc | RB-P | |
| *Llama3.1-8B-Instruct, KV Size = 256* | | | | | | | | | | | | | | | | | |
| HeadKV(GQA) | 30.55 | 32.41 | 48.36 | 53.72 | 30.26 | 45.97 | 23.93 | 23.57 | 23.4 | 57.0 | 90.0 | 40.54 | 6.33 | 93.0 | 58.23 | 48.45 | 44.11 |
| AdaKV(GQA) | 29.45 | 33.7 | 48.96 | 54.48 | 29.59 | 46.09 | 23.58 | 23.13 | 23.27 | 53.0 | 90.79 | 40.49 | 6.33 | 98.5 | 59.47 | 50.31 | 44.45 |
| HeadCompressKV | 29.73 | 36.42 | 51.56 | 54.54 | 30.29 | 45.25 | 24.55 | 23.45 | 23.67 | 66.0 | 89.45 | 40.22 | 6.78 | 99.0 | 59.12 | 49.53 | 45.6 |
| AdaCompressKV | 30.37 | 41.53 | 51.98 | 54.86 | 30.25 | 45.81 | 24.7 | 24.14 | 23.68 | 66.5 | 90.37 | 41.02 | 6.14 | 99.0 | 60.7 | 51.31 | **46.4** |
| *Llama3.1-8B-Instruct, KV Size = 512* | | | | | | | | | | | | | | | | | |
| HeadKV(GQA) | 30.92 | 35.98 | 50.78 | 54.61 | 31.26 | 46.26 | 25.69 | 23.84 | 24.62 | 64.0 | 91.23 | 41.94 | 6.33 | 97.5 | 60.69 | 51.22 | 46.05 |
| AdaKV(GQA) | 30.88 | 38.6 | 52.0 | 54.67 | 30.26 | 46.19 | 25.52 | 23.58 | 24.78 | 61.5 | 91.17 | 42.17 | 6.33 | 99.0 | 61.5 | 52.27 | 46.28 |
| HeadCompressKV | 29.31 | 39.41 | 52.32 | 55.3 | 31.18 | 46.47 | 26.17 | 23.69 | 24.94 | 68.5 | 90.48 | 42.66 | 6.17 | 99.0 | 60.44 | 52.13 | 46.76 |
| AdaCompressKV | 30.52 | 44.25 | 52.55 | 55.06 | 31.14 | 47.09 | 26.63 | 24.1 | 25.1 | 70.0 | 91.77 | 43.66 | 6.08 | 99.5 | 62.54 | 53.76 | **47.73** |
| *Llama3.1-8B-Instruct, KV Size = 1024* | | | | | | | | | | | | | | | | | |
| HeadKV(GQA) | 30.6 | 41.24 | 52.13 | 54.34 | 30.96 | 46.33 | 27.27 | 24.11 | 26.0 | 65.5 | 91.51 | 43.1 | 6.17 | 98.5 | 61.84 | 53.17 | 47.05 |
| AdaKV(GQA) | 31.75 | 44.71 | 52.74 | 54.75 | 31.05 | 46.73 | 27.9 | 23.86 | 26.41 | 69.5 | 92.05 | 42.82 | 6.58 | 99.5 | 62.04 | 54.61 | 47.94 |
| HeadCompressKV | 30.8 | 43.42 | 52.56 | 54.96 | 29.94 | 46.64 | 28.21 | 24.62 | 25.87 | 71.5 | 90.98 | 43.47 | 6.08 | 99.5 | 62.46 | 54.31 | 47.83 |
| AdaCompressKV | 30.68 | 44.73 | 53.54 | 55.25 | 30.9 | 46.95 | 29.0 | 24.59 | 26.5 | 71.0 | 91.79 | 43.44 | 6.5 | 99.5 | 62.99 | 55.86 | **48.33** |
| *Mistral-7B-Instruct-v0.3, KV Size = 128* | | | | | | | | | | | | | | | | | |
| HeadKV(GQA) | 27.55 | 27.82 | 47.79 | 47.85 | 26.93 | 36.83 | 20.63 | 21.66 | 20.12 | 62.0 | 89.39 | 43.28 | 5.5 | 90.5 | 53.89 | 51.48 | 42.08 |
| AdaKV(GQA) | 25.61 | 24.66 | 45.24 | 47.07 | 24.36 | 34.95 | 20.45 | 21.08 | 19.84 | 62.0 | 88.86 | 42.06 | 6.5 | 90.0 | 52.53 | 49.34 | 40.91 |
| HeadCompressKV | 28.49 | 32.0 | 50.69 | 47.43 | 27.74 | 37.77 | 21.47 | 21.99 | 20.64 | 72.0 | 89.69 | 42.55 | 5.5 | 93.5 | 52.41 | 50.34 | 43.39 |
| AdaCompressKV | 28.02 | 30.94 | 53.07 | 49.99 | 26.65 | 36.91 | 21.4 | 22.41 | 21.32 | 72.0 | 89.64 | 42.95 | 6.0 | 90.0 | 52.89 | 50.71 | **43.43** |
| *Mistral-7B-Instruct-v0.3, KV Size = 256* | | | | | | | | | | | | | | | | | |
| HeadKV(GQA) | 29.15 | 31.66 | 51.03 | 49.09 | 24.88 | 36.64 | 22.46 | 22.2 | 22.97 | 70.5 | 89.57 | 43.86 | 5.5 | 94.0 | 56.77 | 55.39 | 44.1 |
| AdaKV(GQA) | 29.08 | 27.93 | 49.69 | 49.4 | 26.07 | 36.34 | 22.55 | 21.72 | 22.14 | 70.5 | 89.69 | 44.09 | 6.5 | 94.0 | 56.21 | 54.05 | 43.75 |
| HeadCompressKV | 29.61 | 33.87 | 52.4 | 49.26 | 28.53 | 37.17 | 22.87 | 23.1 | 23.03 | 76.0 | 89.44 | 44.18 | 5.0 | 96.5 | 55.73 | 55.28 | **45.12** |
| AdaCompressKV | 30.05 | 33.38 | 53.38 | 49.23 | 27.59 | 38.04 | 23.47 | 22.87 | 23.27 | 74.5 | 89.33 | 43.89 | 4.0 | 96.0 | 56.15 | 55.24 | 45.02 |
| *Mistral-7B-Instruct-v0.3, KV Size = 512* | | | | | | | | | | | | | | | | | |
| HeadKV(GQA) | 29.38 | 33.75 | 53.24 | 50.45 | 27.24 | 37.44 | 24.5 | 23.41 | 24.72 | 75.0 | 89.52 | 45.52 | 6.0 | 96.0 | 57.68 | 57.9 | 45.73 |
| AdaKV(GQA) | 29.04 | 32.7 | 51.16 | 49.56 | 25.56 | 37.03 | 24.17 | 22.69 | 24.11 | 73.5 | 89.49 | 45.06 | 6.5 | 95.0 | 57.98 | 57.48 | 45.06 |
| HeadCompressKV | 29.06 | 36.55 | 53.31 | 50.81 | 28.93 | 39.18 | 25.22 | 23.93 | 24.84 | 76.0 | 89.19 | 44.96 | 4.5 | 96.0 | 57.31 | 57.85 | 46.1 |
| AdaCompressKV | 30.5 | 36.74 | 53.75 | 49.39 | 27.62 | 38.16 | 26.01 | 24.19 | 25.13 | 76.0 | 89.49 | 44.71 | 5.5 | 96.0 | 58.04 | 57.52 | **46.17** |
| *Mistral-7B-Instruct-v0.3, KV Size = 1024* | | | | | | | | | | | | | | | | | |
| HeadKV(GQA) | 29.62 | 37.34 | 53.09 | 50.31 | 26.67 | 38.28 | 26.44 | 23.52 | 26.11 | 76.0 | 89.35 | 45.68 | 5.0 | 97.0 | 58.54 | 59.56 | 46.41 |
| AdaKV(GQA) | 29.73 | 37.04 | 53.07 | 50.01 | 26.32 | 37.54 | 26.27 | 23.97 | 26.12 | 75.5 | 89.49 | 46.25 | 6.5 | 96.5 | 58.62 | 59.16 | 46.38 |
| HeadCompressKV | 29.89 | 38.07 | 53.62 | 48.92 | 29.08 | 38.69 | 27.62 | 24.36 | 26.31 | 76.0 | 89.46 | 45.96 | 5.5 | 96.5 | 58.55 | 59.96 | 46.78 |
| AdaCompressKV | 29.83 | 38.86 | 53.1 | 49.18 | 28.56 | 39.06 | 28.06 | 25.1 | 26.82 | 76.0 | 89.24 | 45.66 | 5.5 | 97.0 | 58.17 | 60.26 | **46.9** |

Table 6: Prefilling-stage accelerators and their integration with CompressKV on LongBench. Combined variants maintain accuracy comparable to prefilling-only baselines while enabling decoding-stage memory reduction, confirming orthogonality.

| Method | Single-Document QA | | | Multi-Document QA | | | Summarization | | | Few-shot Learning | | | Synthetic | | Code | | Avg. |
|---|---|---|---|---|---|---|---|---|---|---|---|---|---|---|---|---|---|
| | NrtvQA | Qasper | MF-en | HotpotQA | Musique | 2WikiMQA | GovReport | QMSum | MultiNews | TREC | TriviaQA | SAMSum | PCount | PR-en | Lcc | RB-P | |
| *Mistral-7B-Instruct-v0.3, KV Size = 2048* | | | | | | | | | | | | | | | | | |
| CompressKV | 29.4 | 40.92 | 53.52 | 50.36 | 28.76 | 39.45 | 31.26 | 25.17 | 27.35 | 76.0 | 88.64 | 47.09 | 5.5 | 97.5 | 59.21 | 60.72 | 47.55 |
| MInference | 29.34 | 41.81 | 53.8 | 49.88 | 28.9 | 40.16 | 34.96 | 25.69 | 27.82 | 76.0 | 88.59 | 47.63 | 5.5 | 98.0 | 59.47 | 61.27 | 48.05 |
| MInference+CompressKV | 27.74 | 40.49 | 53.69 | 51.37 | 28.86 | 39.87 | 31.2 | 25.45 | 27.32 | 76.0 | 88.89 | 47.53 | 5.5 | 98.0 | 59.17 | 61.01 | 47.63 |
| XAttention | 27.65 | 35.79 | 53.64 | 51.89 | 27.91 | 39.11 | 34.46 | 24.98 | 27.25 | 74.5 | 89.0 | 47.3 | 6.5 | 89.5 | 59.49 | 61.21 | 46.89 |
| XAttention+CompressKV | 27.51 | 34.77 | 52.91 | 52.27 | 27.4 | 38.73 | 30.67 | 24.54 | 27.25 | 74.5 | 89.27 | 46.43 | 6.5 | 90.5 | 59.27 | 61.41 | 46.5 |

Table 7: Details Performance comparison of CompressKV with StreamingLLM, SnapKV, Pyra-midKV, CAKE, HeadKV, AdaKV and FullKV on LongBench for Llama-3.1-8B-Instruct.

| Method | Single-Document QA | | | Multi-Document QA | | | Summarization | | | Few-shot Learning | | | Synthetic | | Code | | Avg. |
|---|---|---|---|---|---|---|---|---|---|---|---|---|---|---|---|---|---|
| | NrtvQA | Qasper | MF-en | HotpotQA | Musique | 2WikiMQA | GovReport | QMSum | MultiNews | TREC | TriviaQA | SAMSum | PCount | PR-en | Lcc | RB-P | |
| Llama3.1-8B-Instruct, KV Size = Full | | | | | | | | | | | | | | | | | |
| FullKV | 30.97 | 45.49 | 53.78 | 54.76 | 31.42 | 47.13 | 34.9 | 25.28 | 27.48 | 73.0 | 91.65 | 43.8 | 6.0 | 99.5 | 63.38 | 56.73 | 49.08 |
| Llama3.1-8B-Instruct, KV Size = 128 | | | | | | | | | | | | | | | | | |
| StreamingLLM | 23.99 | 21.68 | 30.86 | 42.58 | 20.78 | 25.03 | 21.14 | 15.69 | 20.93 | 43.5 | 71.07 | 17.24 | 6.12 | 84.33 | 37.92 | 33.66 | 32.28 |
| SnapKV | 27.53 | 28.86 | 48.14 | 54.1 | 28.94 | 45.66 | 21.32 | 22.6 | 20.95 | 51.5 | 90.25 | 39.44 | 6.3 | 90.0 | 57.21 | 48.5 | 42.58 |
| PyramidKV | 27.41 | 27.59 | 46.88 | 54.81 | 28.08 | 45.32 | 21.34 | 22.48 | 20.77 | 52.0 | 87.86 | 38.11 | 6.5 | 94.0 | 53.47 | 45.77 | 42.02 |
| CAKE | 31.19 | 35.82 | 51.36 | 53.66 | 30.1 | 43.65 | 23.58 | 23.22 | 22.45 | 60.5 | 90.32 | 40.54 | 6.33 | 99.0 | 57.06 | 48.59 | 44.84 |
| HeadKV(GQA) | 29.51 | 32.23 | 47.49 | 53.44 | 29.1 | 44.89 | 22.11 | 22.59 | 21.74 | 48.5 | 89.3 | 37.99 | 6.25 | 97.0 | 54.93 | 45.46 | 42.66 |
| AdaKV(GQA) | 27.74 | 30.4 | 48.35 | 54.63 | 29.32 | 45.76 | 21.01 | 22.08 | 20.79 | 47.0 | 88.43 | 39.5 | 6.5 | 89.0 | 55.76 | 46.58 | 42.05 |
| CompressKV | 30.43 | 35.74 | 51.12 | 53.8 | 31.09 | 46.13 | 22.58 | 23.42 | 22.05 | 65.5 | 87.56 | 39.43 | 6.28 | 99.0 | 57.3 | 50.1 | **45.1** |
| Llama3.1-8B-Instruct, KV Size = 256 | | | | | | | | | | | | | | | | | |
| StreamingLLM | 23.35 | 22.26 | 33.94 | 42.8 | 21.91 | 24.47 | 23.35 | 16.28 | 23.85 | 54.5 | 70.66 | 17.65 | 6.12 | 87.99 | 39.64 | 34.01 | 33.92 |
| SnapKV | 30.41 | 36.24 | 49.88 | 54.06 | 30.69 | 45.95 | 23.64 | 22.99 | 23.6 | 58.0 | 91.29 | 40.92 | 6.25 | 99.0 | 59.68 | 50.74 | 45.21 |
| PyramidKV | 29.02 | 33.4 | 49.43 | 54.79 | 29.21 | 46.24 | 23.45 | 22.87 | 22.81 | 58.0 | 89.41 | 39.8 | 6.25 | 98.5 | 57.42 | 49.16 | 44.36 |
| CAKE | 31.31 | 40.18 | 51.54 | 54.48 | 29.66 | 45.75 | 25.22 | 23.84 | 24.09 | 64.5 | 91.49 | 42.06 | 6.14 | 99.5 | 59.6 | 51.51 | 46.3 |
| HeadKV(GQA) | 30.55 | 32.41 | 48.36 | 53.72 | 30.26 | 45.97 | 23.93 | 23.57 | 23.4 | 57.0 | 90.0 | 40.54 | 6.33 | 93.0 | 58.23 | 48.45 | 44.11 |
| AdaKV(GQA) | 29.45 | 33.7 | 48.96 | 54.48 | 29.59 | 46.09 | 23.58 | 23.13 | 23.27 | 53.0 | 90.79 | 40.49 | 6.33 | 98.5 | 59.47 | 50.31 | 44.45 |
| CompressKV | 30.68 | 42.58 | 52.27 | 54.37 | 30.43 | 46.46 | 24.86 | 23.79 | 24.13 | 68.5 | 89.97 | 41.08 | 6.14 | 99.5 | 60.63 | 51.94 | **46.71** |
| Llama3.1-8B-Instruct, KV Size = 512 | | | | | | | | | | | | | | | | | |
| StreamingLLM | 24.54 | 25.69 | 36.97 | 42.42 | 22.31 | 25.6 | 26.08 | 17.65 | 25.76 | 61.0 | 71.94 | 18.6 | 6.5 | 87.71 | 41.39 | 35.22 | 35.59 |
| SnapKV | 30.62 | 40.05 | 52.43 | 54.74 | 30.93 | 46.39 | 25.52 | 23.42 | 25.16 | 65.0 | 91.21 | 42.07 | 6.03 | 99.0 | 61.84 | 52.91 | 46.71 |
| PyramidKV | 30.43 | 39.81 | 52.98 | 54.47 | 31.06 | 46.31 | 25.35 | 23.66 | 24.74 | 64.5 | 91.84 | 41.93 | 6.08 | 99.5 | 59.97 | 51.19 | 46.49 |
| CAKE | 30.47 | 43.19 | 51.92 | 54.24 | 30.41 | 45.5 | 27.04 | 24.21 | 25.15 | 70.0 | 91.46 | 42.28 | 6.33 | 99.5 | 61.55 | 52.68 | 47.25 |
| HeadKV(GQA) | 30.92 | 35.98 | 50.78 | 54.61 | 31.26 | 46.26 | 25.69 | 23.84 | 24.62 | 64.0 | 91.23 | 41.94 | 6.33 | 97.5 | 60.69 | 51.22 | 46.05 |
| AdaKV(GQA) | 30.88 | 38.6 | 52.0 | 54.67 | 30.26 | 46.19 | 25.52 | 23.58 | 24.78 | 61.5 | 91.17 | 42.17 | 6.33 | 99.0 | 61.5 | 52.27 | 46.28 |
| CompressKV | 30.8 | 44.41 | 53.13 | 55.32 | 30.67 | 46.89 | 27.0 | 23.92 | 25.17 | 70.5 | 92.0 | 43.5 | 6.58 | 99.5 | 62.1 | 53.01 | **47.78** |
| Llama3.1-8B-Instruct, KV Size = 1024 | | | | | | | | | | | | | | | | | |
| StreamingLLM | 23.55 | 28.99 | 43.17 | 42.5 | 21.61 | 28.37 | 28.26 | 19.12 | 26.37 | 68.0 | 74.02 | 19.41 | 6.19 | 82.58 | 43.65 | 35.49 | 36.95 |
| SnapKV | 31.32 | 44.62 | 52.51 | 54.65 | 30.24 | 46.8 | 28.14 | 24.12 | 26.36 | 69.0 | 92.05 | 42.67 | 6.12 | 99.5 | 62.06 | 54.99 | 47.82 |
| PyramidKV | 31.06 | 44.09 | 53.25 | 54.25 | 30.45 | 46.87 | 27.44 | 23.46 | 26.31 | 70.0 | 91.93 | 42.91 | 6.08 | 99.5 | 62.03 | 52.75 | 47.65 |
| CAKE | 30.61 | 44.64 | 52.18 | 54.89 | 30.44 | 46.14 | 29.1 | 24.28 | 26.33 | 71.0 | 91.89 | 42.81 | 6.17 | 99.5 | 62.05 | 55.47 | 47.97 |
| HeadKV(GQA) | 30.6 | 41.24 | 52.13 | 54.34 | 30.96 | 46.33 | 27.27 | 24.11 | 26.0 | 65.5 | 91.51 | 43.1 | 6.17 | 98.5 | 61.84 | 53.17 | 47.05 |
| AdaKV(GQA) | 31.75 | 44.71 | 52.74 | 54.75 | 31.05 | 46.73 | 27.9 | 23.86 | 26.41 | 69.5 | 92.05 | 42.82 | 6.58 | 99.5 | 62.04 | 54.61 | 47.94 |
| CompressKV | 30.27 | 45.18 | 53.44 | 55.25 | 30.83 | 46.57 | 29.06 | 24.34 | 26.48 | 71.0 | 91.79 | 43.37 | 6.0 | 99.5 | 63.14 | 55.62 | **48.24** |
| Llama3.1-8B-Instruct, KV Size = 2048 | | | | | | | | | | | | | | | | | |
| StreamingLLM | 25.2 | 37.54 | 48.31 | 44.74 | 23.09 | 31.47 | 30.61 | 19.51 | 27.15 | 69.0 | 79.5 | 21.21 | 5.73 | 71.75 | 51.99 | 37.02 | 38.99 |
| SnapKV | 30.61 | 45.01 | 53.22 | 55.2 | 30.62 | 46.21 | 30.57 | 24.1 | 27.2 | 71.5 | 91.65 | 43.91 | 6.0 | 99.5 | 63.27 | 56.9 | 48.47 |
| PyramidKV | 30.99 | 44.83 | 52.73 | 54.47 | 31.33 | 46.76 | 29.64 | 23.92 | 27.22 | 72.5 | 91.92 | 43.04 | 6.55 | 99.5 | 63.12 | 54.89 | 48.34 |
| CAKE | 30.56 | 44.58 | 52.54 | 54.82 | 30.36 | 46.43 | 30.38 | 24.72 | 27.24 | 71.5 | 91.48 | 43.04 | 6.17 | 99.5 | 63.05 | 55.84 | 48.26 |
| HeadKV(GQA) | 30.7 | 44.14 | 53.2 | 54.78 | 30.65 | 46.42 | 29.27 | 23.94 | 26.98 | 70.5 | 91.81 | 43.81 | 6.0 | 99.5 | 63.01 | 55.34 | 48.13 |
| AdaKV(GQA) | 31.08 | 45.42 | 52.8 | 54.81 | 30.46 | 46.19 | 30.14 | 24.57 | 27.22 | 70.5 | 91.66 | 43.52 | 6.0 | 99.5 | 63.14 | 56.56 | 48.35 |
| CompressKV | 30.83 | 45.59 | 53.86 | 55.04 | 31.05 | 46.43 | 31.62 | 24.86 | 27.15 | 72.0 | 91.65 | 43.67 | 6.0 | 99.5 | 63.3 | 56.73 | **48.7** |

Table 8: Details Performance comparison of CompressKV with StreamingLLM, SnapKV, Pyra-midKV, CAKE, HeadKV, AdaKV and FullKV on LongBench for Mistral-7B-Instruct-v0.3.

| Method | Single-Document QA | | | Multi-Document QA | | | Summarization | | | Few-shot Learning | | | Synthetic | | Code | | Avg. |
|---|---|---|---|---|---|---|---|---|---|---|---|---|---|---|---|---|---|
| | NrtvQA | Qasper | MF-en | HotpotQA | Musique | 2WikiMQA | GovReport | QMSum | MultiNews | TREC | TriviaQA | SAMSum | PCount | PR-en | Lcc | RB-P | |
| Mistral-7B-Instruct-v0.3, KV Size = Full | | | | | | | | | | | | | | | | | |
| FullKV | 29.07 | 41.54 | 52.88 | 49.37 | 28.58 | 39.01 | 35.07 | 25.71 | 27.73 | 76.0 | 88.59 | 47.51 | 6.0 | 98.0 | 59.35 | 60.7 | 47.82 |
| Mistral-7B-Instruct-v0.3, KV Size = 128 | | | | | | | | | | | | | | | | | |
| StreamingLLM | 21.06 | 20.03 | 29.31 | 36.98 | 16.77 | 24.14 | 18.97 | 17.29 | 20.19 | 46.5 | 73.66 | 18.08 | 4.5 | 67.75 | 29.45 | 32.86 | 29.85 |
| SnapKV | 25.95 | 25.46 | 45.79 | 47.79 | 25.27 | 36.47 | 20.63 | 21.02 | 19.96 | 58.5 | 89.16 | 41.94 | 5.0 | 86.0 | 53.3 | 50.39 | 40.79 |
| PyramidKV | 24.95 | 24.43 | 46.35 | 45.77 | 24.74 | 35.45 | 20.93 | 21.26 | 20.07 | 58.0 | 88.63 | 40.82 | 6.0 | 89.0 | 50.92 | 47.34 | 40.29 |
| CAKE | 28.15 | 28.81 | 50.69 | 48.27 | 26.99 | 37.8 | 22.71 | 23.37 | 22.0 | 63.0 | 89.31 | 42.95 | 6.5 | 93.0 | 53.25 | 51.84 | 43.04 |
| HeadKV(GQA) | 27.55 | 27.82 | 47.79 | 47.85 | 26.93 | 36.83 | 20.63 | 21.66 | 20.12 | 62.0 | 89.39 | 43.28 | 5.5 | 90.5 | 53.89 | 51.48 | 42.08 |
| AdaKV(GQA) | 25.61 | 24.66 | 45.24 | 47.07 | 24.36 | 34.95 | 20.45 | 21.08 | 19.84 | 62.0 | 88.86 | 42.06 | 6.5 | 90.0 | 52.53 | 49.34 | 40.91 |
| CompressKV | 28.06 | 31.74 | 52.62 | 49.75 | 25.99 | 37.1 | 21.76 | 22.53 | 21.58 | 69.0 | 89.89 | 43.35 | 6.0 | 93.5 | 52.74 | 51.27 | **43.55** |
| Mistral-7B-Instruct-v0.3, KV Size = 256 | | | | | | | | | | | | | | | | | |
| StreamingLLM | 21.89 | 21.37 | 32.51 | 36.98 | 16.78 | 25.43 | 21.85 | 16.89 | 23.55 | 57.5 | 72.38 | 18.24 | 4.0 | 65.0 | 31.63 | 33.53 | 31.22 |
| SnapKV | 27.49 | 29.18 | 48.92 | 48.0 | 26.47 | 36.78 | 22.46 | 22.04 | 22.54 | 70.0 | 89.44 | 43.72 | 6.0 | 96.0 | 56.69 | 54.49 | 43.76 |
| PyramidKV | 27.71 | 28.23 | 48.25 | 48.61 | 25.62 | 36.18 | 22.1 | 21.77 | 21.8 | 70.0 | 89.33 | 43.65 | 6.0 | 93.5 | 55.13 | 51.06 | 43.06 |
| CAKE | 27.46 | 33.09 | 54.32 | 48.32 | 27.08 | 37.8 | 24.62 | 23.61 | 23.85 | 70.0 | 89.33 | 44.09 | 4.5 | 95.5 | 55.76 | 56.35 | 44.73 |
| HeadKV(GQA) | 29.15 | 31.66 | 51.03 | 49.09 | 24.88 | 36.64 | 22.46 | 22.2 | 22.97 | 70.5 | 89.57 | 43.86 | 5.5 | 94.0 | 56.77 | 55.39 | 44.1 |
| AdaKV(GQA) | 29.08 | 27.93 | 49.69 | 49.4 | 26.07 | 36.34 | 22.55 | 21.72 | 22.14 | 70.5 | 89.69 | 44.09 | 6.5 | 94.0 | 56.21 | 54.05 | 43.75 |
| CompressKV | 29.86 | 35.32 | 52.84 | 49.77 | 27.64 | 38.04 | 24.05 | 23.37 | 23.26 | 75.5 | 89.16 | 45.31 | 4.5 | 96.5 | 56.47 | 55.31 | **45.43** |
| Mistral-7B-Instruct-v0.3, KV Size = 512 | | | | | | | | | | | | | | | | | |
| StreamingLLM | 21.13 | 22.16 | 34.59 | 37.86 | 16.08 | 26.1 | 25.19 | 18.47 | 26.53 | 65.5 | 71.51 | 18.13 | 3.25 | 68.0 | 34.13 | 33.7 | 32.65 |
| SnapKV | 29.43 | 34.49 | 52.68 | 50.1 | 26.08 | 37.04 | 23.97 | 22.94 | 24.06 | 73.0 | 89.55 | 45.23 | 5.5 | 96.5 | 57.94 | 57.54 | 45.38 |
| PyramidKV | 27.87 | 33.32 | 51.53 | 49.63 | 25.42 | 36.91 | 24.0 | 22.76 | 24.14 | 72.0 | 90.07 | 44.49 | 5.5 | 96.5 | 57.33 | 55.6 | 44.82 |
| CAKE | 29.5 | 35.89 | 53.79 | 49.74 | 27.13 | 39.32 | 26.7 | 23.81 | 25.44 | 72.5 | 89.27 | 45.78 | 5.5 | 97.5 | 57.35 | 57.71 | 46.06 |
| HeadKV(GQA) | 29.38 | 33.75 | 53.24 | 50.45 | 27.24 | 37.44 | 24.5 | 23.41 | 24.72 | 75.0 | 89.52 | 45.52 | 6.0 | 96.0 | 57.68 | 57.9 | 45.73 |
| AdaKV(GQA) | 29.04 | 32.7 | 51.16 | 49.56 | 25.56 | 37.03 | 24.17 | 22.69 | 24.11 | 73.5 | 89.49 | 45.06 | 6.5 | 95.0 | 57.98 | 57.48 | 45.06 |
| CompressKV | 29.91 | 37.01 | 54.3 | 49.31 | 27.93 | 38.1 | 26.35 | 23.97 | 24.99 | 76.0 | 89.49 | 44.81 | 7.0 | 96.0 | 58.41 | 58.65 | **46.39** |
| Mistral-7B-Instruct-v0.3, KV Size = 1024 | | | | | | | | | | | | | | | | | |
| StreamingLLM | 22.37 | 28.03 | 41.21 | 38.0 | 17.05 | 26.95 | 27.75 | 19.87 | 27.13 | 71.5 | 70.34 | 19.02 | 5.37 | 68.5 | 37.95 | 34.57 | 34.73 |
| SnapKV | 29.82 | 36.49 | 52.64 | 50.33 | 26.88 | 38.53 | 26.39 | 23.94 | 25.84 | 75.0 | 89.24 | 46.73 | 6.5 | 97.0 | 58.57 | 59.86 | 46.48 |
| PyramidKV | 28.14 | 35.83 | 54.28 | 49.88 | 25.68 | 38.31 | 25.91 | 23.82 | 25.42 | 74.5 | 89.86 | 46.17 | 5.0 | 97.5 | 57.72 | 57.36 | 45.96 |
| CAKE | 29.21 | 36.86 | 53.2 | 48.69 | 27.62 | 38.78 | 28.94 | 24.54 | 26.98 | 74.0 | 88.94 | 46.94 | 5.0 | 98.0 | 58.88 | 59.91 | 46.66 |
| HeadKV(GQA) | 29.62 | 37.34 | 53.09 | 50.31 | 26.67 | 38.28 | 26.44 | 23.52 | 26.11 | 76.0 | 89.35 | 45.68 | 5.0 | 97.0 | 58.54 | 59.56 | 46.41 |
| AdaKV(GQA) | 29.73 | 37.04 | 53.07 | 50.01 | 26.32 | 37.54 | 26.27 | 23.97 | 26.12 | 75.5 | 89.49 | 46.25 | 6.5 | 96.5 | 58.62 | 59.16 | 46.38 |
| CompressKV | 29.75 | 38.88 | 52.81 | 49.71 | 28.48 | 39.04 | 28.26 | 24.95 | 26.88 | 76.0 | 89.24 | 46.16 | 5.5 | 97.0 | 58.65 | 60.05 | **46.96** |
| Mistral-7B-Instruct-v0.3, KV Size = 2048 | | | | | | | | | | | | | | | | | |
| StreamingLLM | 23.53 | 31.86 | 47.11 | 37.98 | 18.91 | 29.27 | 30.10 | 20.26 | 27.18 | 73.0 | 69.97 | 19.01 | 4.75 | 72.25 | 42.87 | 36.07 | 36.51 |
| SnapKV | 30.54 | 39.94 | 53.16 | 50.18 | 27.29 | 38.49 | 28.47 | 24.06 | 27.46 | 76.0 | 88.48 | 46.06 | 5.0 | 98.0 | 59.38 | 60.35 | 47.05 |
| PyramidKV | 29.21 | 39.62 | 52.78 | 50.47 | 27.71 | 37.88 | 28.46 | 24.27 | 27.38 | 75.0 | 89.86 | 46.32 | 5.0 | 98.0 | 58.60 | 59.05 | 46.85 |
| CAKE | 29.91 | 40.06 | 53.56 | 48.11 | 28.06 | 38.78 | 31.05 | 24.90 | 27.53 | 75.0 | 88.64 | 47.08 | 5.5 | 97.5 | 59.56 | 60.35 | 47.22 |
| HeadKV(GQA) | 30.42 | 39.86 | 53.21 | 49.96 | 27.77 | 38.13 | 28.71 | 24.83 | 27.26 | 76.0 | 89.49 | 45.96 | 5.5 | 98.0 | 59.65 | 60.51 | 47.20 |
| AdaKV(GQA) | 30.41 | 40.25 | 52.93 | 50.31 | 27.67 | 38.71 | 28.96 | 24.31 | 27.26 | 76.0 | 88.48 | 46.25 | 5.5 | 98.0 | 59.17 | 60.33 | 47.16 |
| CompressKV | 29.40 | 40.92 | 53.52 | 50.36 | 28.76 | 39.45 | 31.26 | 25.17 | 27.35 | 76.0 | 88.64 | 47.09 | 5.5 | 97.5 | 59.21 | 60.72 | **47.55** |

Table 9: Details Performance comparison of CompressKV with StreamingLLM, SnapKV, PyramidKV, CAKE, HeadKV, AdaKV and FullKV on LongBench for Qwen2.5-7B-Instruct.

| Method | Single-Document QA | | | Multi-Document QA | | | Summarization | | | Few-shot Learning | | | Synthetic | | Code | | Avg. |
|---|---|---|---|---|---|---|---|---|---|---|---|---|---|---|---|---|---|
| | NrtvQA | Qasper | MF-en | HotpotQA | Musique | 2WikiMQA | GovReport | QMSum | MultiNews | TREC | TriviaQA | SAMSum | PCount | PR-en | Lcc | RB-P | |
| Qwen2.5-7B-Instruct, KV Size = Full | | | | | | | | | | | | | | | | | |
| FullKV | 28.14 | 43.76 | 52.61 | 57.7 | 29.67 | 47.19 | 31.89 | 23.61 | 23.91 | 71.5 | 89.97 | 46.16 | 8.5 | 100.0 | 60.56 | 66.83 | 48.88 |
| Qwen2.5-7B-Instruct, KV Size = 128 | | | | | | | | | | | | | | | | | |
| StreamingLLM | 16.52 | 19.94 | 25.07 | 35.1 | 14.16 | 20.74 | 19.42 | 14.54 | 16.98 | 45.0 | 62.68 | 17.9 | 8.5 | 29.33 | 31.34 | 32.73 | 25.62 |
| SnapKV | 24.57 | 28.43 | 46.92 | 51.78 | 26.54 | 43.94 | 19.64 | 19.99 | 16.54 | 46.5 | 87.26 | 40.39 | 9.0 | 99.0 | 49.62 | 53.45 | 41.47 |
| PyramidKV | 21.59 | 27.34 | 42.46 | 50.67 | 23.19 | 43.56 | 17.72 | 18.81 | 14.14 | 43.5 | 86.34 | 38.92 | 9.0 | 83.5 | 44.54 | 46.8 | 38.25 |
| CAKE | 26.95 | 33.29 | 47.14 | 53.11 | 25.36 | 43.98 | 20.99 | 20.81 | 18.14 | 47.5 | 82.47 | 42.82 | 8.5 | 97.5 | 50.97 | 53.22 | 42.05 |
| HeadKV(GQA) | 26.52 | 31.25 | 47.66 | 53.2 | 27.54 | 43.28 | 19.67 | 19.99 | 16.86 | 48.5 | 88.34 | 41.22 | 8.5 | 98.0 | 50.74 | 53.75 | 42.19 |
| AdaKV(GQA) | 24.78 | 29.21 | 44.99 | 52.57 | 26.91 | 43.59 | 19.57 | 20.06 | 16.57 | 45.5 | 87.94 | 41.2 | 9.0 | 97.5 | 49.63 | 52.37 | 41.34 |
| CompressKV | 27.3 | 32.56 | 48.43 | 53.94 | 26.43 | 43.2 | 20.35 | 20.19 | 17.0 | 56.0 | 85.63 | 42.23 | 8.5 | 97.0 | 50.62 | 52.45 | **42.61** |
| Qwen2.5-7B-Instruct, KV Size = 256 | | | | | | | | | | | | | | | | | |
| StreamingLLM | 18.04 | 21.45 | 27.37 | 35.88 | 14.17 | 19.45 | 22.23 | 15.54 | 20.77 | 52.5 | 62.65 | 17.86 | 8.5 | 25.33 | 33.76 | 33.51 | 26.81 |
| SnapKV | 27.32 | 35.61 | 49.31 | 54.66 | 27.11 | 43.67 | 21.93 | 20.88 | 19.33 | 53.0 | 88.04 | 43.2 | 8.5 | 98.0 | 55.08 | 56.46 | 43.88 |
| PyramidKV | 25.05 | 33.19 | 47.72 | 52.62 | 25.28 | 44.05 | 20.14 | 19.92 | 16.64 | 50.5 | 87.76 | 40.71 | 8.5 | 94.0 | 50.36 | 51.77 | 41.76 |
| CAKE | 26.56 | 36.42 | 49.88 | 54.25 | 27.56 | 45.93 | 23.22 | 20.82 | 19.82 | 53.0 | 86.08 | 43.59 | 8.5 | 97.0 | 54.44 | 57.81 | 44.05 |
| HeadKV(GQA) | 27.62 | 36.57 | 49.71 | 54.96 | 26.82 | 44.83 | 22.51 | 20.43 | 19.5 | 56.5 | 89.12 | 43.43 | 8.5 | 98.5 | 55.63 | 58.33 | 44.56 |
| AdaKV(GQA) | 27.19 | 35.54 | 48.15 | 54.34 | 27.8 | 44.7 | 21.74 | 20.44 | 18.99 | 53.5 | 87.65 | 42.5 | 8.5 | 98.0 | 55.04 | 56.65 | 43.8 |
| CompressKV | 28.94 | 36.39 | 48.85 | 54.45 | 26.65 | 44.13 | 22.37 | 20.96 | 19.02 | 65.5 | 87.5 | 42.08 | 8.5 | 98.5 | 53.72 | 57.46 | **44.69** |
| Qwen2.5-7B-Instruct, KV Size = 512 | | | | | | | | | | | | | | | | | |
| StreamingLLM | 18.99 | 23.62 | 32.66 | 36.5 | 15.14 | 22.27 | 25.27 | 16.79 | 22.57 | 59.5 | 61.28 | 18.18 | 8.5 | 14.0 | 35.46 | 34.79 | 27.85 |
| SnapKV | 28.52 | 38.99 | 49.77 | 56.46 | 27.73 | 44.58 | 24.09 | 21.06 | 20.86 | 64.0 | 88.91 | 43.92 | 8.5 | 99.0 | 57.38 | 61.03 | 45.93 |
| PyramidKV | 29.66 | 36.78 | 49.29 | 55.8 | 26.6 | 44.96 | 22.54 | 20.47 | 18.35 | 64.5 | 88.46 | 42.23 | 8.5 | 99.0 | 53.21 | 56.39 | 44.8 |
| CAKE | 29.6 | 39.9 | 50.35 | 55.92 | 29.3 | 44.01 | 25.56 | 21.75 | 21.35 | 62.5 | 87.95 | 44.44 | 8.5 | 99.5 | 56.9 | 60.9 | 46.15 |
| HeadKV(GQA) | 28.13 | 39.26 | 50.52 | 56.47 | 26.68 | 44.57 | 24.9 | 21.4 | 21.17 | 65.0 | 89.2 | 43.56 | 8.5 | 99.0 | 57.53 | 61.87 | 46.11 |
| AdaKV(GQA) | 29.18 | 39.17 | 49.16 | 55.7 | 26.94 | 43.83 | 24.08 | 21.02 | 20.91 | 64.0 | 89.29 | 43.32 | 8.5 | 99.0 | 57.24 | 60.68 | 45.75 |
| CompressKV | 29.95 | 41.09 | 51.54 | 56.31 | 28.74 | 45.05 | 24.7 | 21.62 | 20.99 | 68.0 | 87.95 | 43.88 | 8.5 | 98.5 | 57.1 | 61.38 | **46.58** |
| Qwen2.5-7B-Instruct, KV Size = 1024 | | | | | | | | | | | | | | | | | |
| StreamingLLM | 20.68 | 30.34 | 41.03 | 37.96 | 16.35 | 25.71 | 27.04 | 17.89 | 23.39 | 67.0 | 61.54 | 18.18 | 8.5 | 12.5 | 39.15 | 36.4 | 30.23 |
| SnapKV | 28.77 | 40.73 | 51.54 | 56.84 | 28.64 | 45.65 | 26.7 | 21.87 | 22.84 | 69.0 | 89.57 | 44.35 | 8.5 | 99.5 | 59.65 | 64.85 | 47.44 |
| PyramidKV | 29.71 | 39.75 | 52.22 | 56.39 | 26.09 | 45.77 | 24.57 | 20.91 | 21.19 | 68.5 | 89.14 | 44.06 | 8.5 | 99.0 | 58.44 | 60.03 | 46.52 |
| CAKE | 28.88 | 42.2 | 51.66 | 56.66 | 28.96 | 45.25 | 28.0 | 21.98 | 23.09 | 68.5 | 88.73 | 44.85 | 8.5 | 100.0 | 58.95 | 64.15 | 47.52 |
| HeadKV(GQA) | 28.6 | 41.32 | 52.07 | 57.09 | 28.98 | 45.32 | 27.39 | 21.91 | 23.11 | 69.5 | 89.58 | 44.58 | 8.6 | 99.5 | 59.56 | 64.47 | 47.59 |
| AdaKV(GQA) | 28.9 | 40.86 | 52.09 | 56.93 | 26.62 | 45.64 | 26.51 | 21.87 | 22.89 | 69.5 | 89.74 | 44.44 | 8.5 | 99.5 | 59.83 | 63.32 | 47.32 |
| CompressKV | 29.52 | 42.17 | 51.82 | 57.55 | 29.5 | 45.17 | 26.96 | 22.06 | 22.69 | 70.5 | 88.43 | 44.11 | 8.5 | 100.0 | 59.08 | 63.94 | **47.62** |
| Qwen2.5-7B-Instruct, KV Size = 2048 | | | | | | | | | | | | | | | | | |
| StreamingLLM | 21.87 | 36.22 | 46.03 | 38.79 | 17.96 | 31.26 | 28.31 | 18.67 | 23.63 | 68.5 | 65.02 | 19.81 | 8.5 | 17.5 | 47.11 | 37.74 | 32.93 |
| SnapKV | 28.96 | 43.09 | 53.06 | 56.73 | 29.66 | 46.9 | 28.84 | 22.13 | 23.71 | 70.0 | 89.72 | 44.32 | 8.5 | 100.0 | 60.17 | 66.31 | 48.26 |
| PyramidKV | 28.5 | 42.27 | 52.58 | 55.5 | 28.36 | 47.21 | 26.42 | 21.87 | 23.78 | 71.0 | 89.48 | 44.3 | 8.5 | 99.5 | 59.97 | 61.78 | 47.56 |
| CAKE | 28.73 | 42.87 | 52.11 | 56.34 | 29.47 | 46.43 | 29.5 | 22.69 | 23.57 | 70.0 | 89.63 | 45.58 | 8.5 | 100.0 | 59.86 | 66.17 | 48.22 |
| HeadKV(GQA) | 28.43 | 42.99 | 52.73 | 57.08 | 30.0 | 46.68 | 29.58 | 22.47 | 23.85 | 70.5 | 89.72 | 44.88 | 8.5 | 100.0 | 60.04 | 66.55 | **48.38** |
| AdaKV(GQA) | 29.76 | 43.11 | 52.93 | 56.39 | 28.14 | 46.5 | 28.67 | 22.08 | 23.8 | 70.5 | 89.72 | 44.86 | 8.5 | 100.0 | 60.36 | 65.39 | 48.17 |
| CompressKV | 28.48 | 43.24 | 51.83 | 57.17 | 30.54 | 46.99 | 28.98 | 22.53 | 23.43 | 71.0 | 89.84 | 45.15 | 8.5 | 100.0 | 59.65 | 65.46 | 48.30 |

Table 10: Performance comparison of CompressKV with StreamingLLM, SnapKV, PyramidKV, CAKE, HeadKV, AdaKV and FullKV on LongBench for large-scale LLMs(Qwen2.5-14B-Instruct and Qwen2.5-32B-Instruct).

| Method | Single-Document QA | | | Multi-Document QA | | | Summarization | | | Few-shot Learning | | | Synthetic | | Code | | Avg. |
|---|---|---|---|---|---|---|---|---|---|---|---|---|---|---|---|---|---|
| | NrtvQA | Qasper | MF-en | HotpotQA | Musique | 2WikiMQA | GovReport | QMSum | MultiNews | TREC | TriviaQA | SAMSum | PCount | PR-en | Lcc | RB-P | |
| *Qwen2.5-14B-Instruct, KV Size = Full* | | | | | | | | | | | | | | | | | |
| FullKV | 30.11 | 45.21 | 53.79 | 61.99 | 37.43 | 58.21 | 29.53 | 23.41 | 22.07 | 77.0 | 90.69 | 47.89 | 10.13 | 98.67 | 61.34 | 49.33 | 49.8 |
| *Qwen2.5-14B-Instruct, KV Size = 256* | | | | | | | | | | | | | | | | | |
| StreamingLLM | 17.32 | 20.57 | 25.08 | 36.12 | 22.3 | 25.1 | 20.68 | 14.56 | 19.49 | 54.5 | 61.93 | 17.15 | 0.17 | 13.33 | 37.96 | 27.45 | 25.86 |
| SnapKV | 25.03 | 29.88 | 42.89 | 56.9 | 36.21 | 53.83 | 20.44 | 20.3 | 17.61 | 58.5 | 88.01 | 45.27 | 7.27 | 97.5 | 57.14 | 43.5 | 43.77 |
| PyramidKV | 23.2 | 28.44 | 42.36 | 57.22 | 35.56 | 55.12 | 19.45 | 19.56 | 16.11 | 58.5 | 87.22 | 43.88 | 8.1 | 95.75 | 52.76 | 40.08 | 42.71 |
| CAKE | 24.14 | 33.4 | 43.14 | 57.16 | 35.68 | 55.36 | 21.53 | 21.32 | 18.45 | 62.0 | 89.21 | 45.64 | 8.1 | 98.0 | 57.7 | 44.38 | 44.7 |
| HeadKV | 26.21 | 31.4 | 43.11 | 57.08 | 36.11 | 53.94 | 20.9 | 20.75 | 17.73 | 60.5 | 88.51 | 45.97 | 8.88 | 95.58 | 56.55 | 44.21 | 44.21 |
| AdaKV | 25.66 | 30.56 | 42.98 | 54.52 | 36.1 | 53.65 | 20.46 | 20.62 | 17.6 | 57.5 | 88.37 | 44.72 | 8.05 | 97.0 | 57.23 | 43.85 | 43.68 |
| CompressKV | 26.11 | 34.99 | 43.76 | 57.86 | 35.25 | 55.55 | 20.81 | 21.02 | 17.68 | 72.0 | 88.57 | 45.34 | 9.14 | 95.92 | 57.49 | 44.43 | **45.37** |
| *Qwen2.5-14B-Instruct, KV Size = 1024* | | | | | | | | | | | | | | | | | |
| StreamingLLM | 19.52 | 29.65 | 39.87 | 39.67 | 22.74 | 31.34 | 25.47 | 17.08 | 21.61 | 71.5 | 61.71 | 18.67 | 1.6 | 7.92 | 41.83 | 27.27 | 29.84 |
| SnapKV | 27.52 | 43.57 | 50.92 | 59.71 | 36.82 | 56.51 | 24.69 | 21.98 | 20.71 | 75.0 | 90.37 | 47.17 | 8.35 | 98.75 | 60.64 | 48.22 | 48.18 |
| PyramidKV | 27.38 | 42.54 | 51.11 | 59.39 | 36.28 | 56.88 | 23.61 | 21.38 | 20.0 | 75.5 | 89.88 | 46.17 | 8.95 | 98.42 | 59.83 | 45.94 | 47.7 |
| CAKE | 28.82 | 44.37 | 51.75 | 60.1 | 36.86 | 57.64 | 26.07 | 22.45 | 21.1 | 74.0 | 90.14 | 47.0 | 8.3 | 98.42 | 60.84 | 48.5 | 48.52 |
| HeadKV | 27.54 | 44.1 | 51.72 | 59.51 | 36.32 | 57.09 | 25.43 | 22.24 | 21.0 | 75.5 | 90.47 | 47.27 | 7.43 | 98.75 | 61.74 | 48.66 | 48.42 |
| AdaKV | 27.49 | 43.3 | 51.15 | 59.22 | 37.4 | 55.85 | 24.82 | 21.76 | 20.59 | 75.5 | 90.37 | 47.52 | 7.77 | 98.5 | 61.38 | 48.47 | 48.19 |
| CompressKV | 28.96 | 44.26 | 51.63 | 60.49 | 37.23 | 56.92 | 25.42 | 22.61 | 20.83 | 77.0 | 90.32 | 47.32 | 8.91 | 98.42 | 61.14 | 47.61 | **48.69** |
| *Qwen2.5-32B-Instruct, KV Size = Full* | | | | | | | | | | | | | | | | | |
| FullKV | 31.02 | 44.24 | 52.18 | 63.21 | 38.73 | 61.0 | 30.25 | 23.59 | 22.82 | 73.5 | 87.78 | 45.77 | 10.5 | 100.0 | 53.23 | 39.26 | 48.57 |
| *Qwen2.5-32B-Instruct, KV Size = 256* | | | | | | | | | | | | | | | | | |
| StreamingLLM | 16.65 | 21.13 | 23.6 | 37.32 | 20.02 | 26.09 | 20.28 | 14.37 | 20.33 | 53.5 | 62.54 | 16.44 | 10.0 | 9.33 | 31.15 | 21.71 | 25.28 |
| SnapKV | 27.22 | 30.06 | 41.9 | 56.1 | 36.32 | 56.91 | 20.3 | 19.58 | 18.22 | 63.5 | 85.95 | 42.53 | 10.5 | 99.33 | 49.77 | 35.61 | 43.36 |
| PyramidKV | 25.46 | 28.73 | 41.27 | 56.29 | 34.38 | 56.22 | 18.92 | 19.04 | 16.44 | 64.0 | 85.8 | 41.2 | 10.0 | 96.33 | 47.53 | 32.19 | 42.11 |
| CAKE | 28.97 | 34.66 | 44.01 | 57.93 | 34.11 | 58.95 | 21.5 | 20.45 | 19.25 | 67.0 | 84.81 | 42.82 | 10.79 | 99.83 | 50.61 | 36.17 | 44.49 |
| HeadKV | 28.09 | 31.82 | 42.77 | 57.24 | 36.44 | 57.56 | 21.3 | 19.71 | 18.71 | 64.0 | 87.17 | 42.31 | 10.5 | 99.5 | 50.29 | 36.97 | 44.02 |
| AdaKV | 27.21 | 30.98 | 40.9 | 56.57 | 35.92 | 57.04 | 20.59 | 19.33 | 18.27 | 63.0 | 85.0 | 42.46 | 10.07 | 99.33 | 50.36 | 35.74 | 43.3 |
| CompressKV | 27.9 | 33.99 | 44.52 | 58.76 | 35.23 | 58.83 | 21.24 | 20.29 | 18.58 | 70.5 | 87.93 | 42.44 | 11.0 | 99.5 | 50.29 | 34.62 | **44.73** |
| *Qwen2.5-32B-Instruct, KV Size = 1024* | | | | | | | | | | | | | | | | | |
| StreamingLLM | 20.07 | 30.1 | 38.15 | 39.68 | 20.73 | 30.59 | 25.46 | 17.37 | 22.27 | 68.0 | 61.9 | 17.32 | 12.0 | 10.92 | 35.69 | 23.73 | 29.62 |
| SnapKV | 30.13 | 41.12 | 49.47 | 61.84 | 38.39 | 61.02 | 25.18 | 21.0 | 21.77 | 74.0 | 87.84 | 44.42 | 10.1 | 100.0 | 53.32 | 37.84 | 47.34 |
| PyramidKV | 31.48 | 41.21 | 48.95 | 61.16 | 38.85 | 61.12 | 23.15 | 20.29 | 20.62 | 73.5 | 88.37 | 44.23 | 10.5 | 100.0 | 51.67 | 36.66 | 46.98 |
| CAKE | 29.28 | 42.6 | 50.01 | 61.98 | 38.02 | 60.47 | 26.16 | 21.96 | 22.28 | 72.5 | 87.66 | 44.62 | 11.0 | 100.0 | 53.33 | 38.25 | 47.51 |
| HeadKV | 30.48 | 42.59 | 49.41 | 61.66 | 37.9 | 61.58 | 26.37 | 21.38 | 21.99 | 73.5 | 87.84 | 43.78 | 10.5 | 100.0 | 53.05 | 37.96 | 47.5 |
| AdaKV | 29.44 | 41.29 | 48.98 | 61.9 | 38.74 | 61.11 | 25.12 | 21.26 | 21.75 | 74.0 | 87.81 | 44.62 | 10.1 | 100.0 | 53.41 | 37.72 | 47.33 |
| CompressKV | 31.07 | 43.9 | 49.96 | 62.53 | 39.51 | 61.14 | 25.63 | 21.47 | 21.92 | 73.5 | 88.49 | 43.88 | 11.0 | 100.0 | 52.46 | 37.97 | **47.78** |

## G    HEAD VISUALIZATION

In Figures 10, we present a comparison between traditional Retrieval Heads and Semantic Retrieval Heads identified using Mistral-7B-Instruct-v0.3. All scores are L1-normalized across the attention head importance distributions. Unlike traditional methods that require exact top-$k$ attention hits, our approach aggregates scores over entire answer spans, capturing heads that contribute semantically relevant context even when they never achieve top-1 attention for individual tokens. For instance, as shown in Figure 10, layers 0 and 1 of the Mistral model have zero scores for all heads using the traditional method, whereas our approach successfully identifies heads of lower yet meaningful importance.

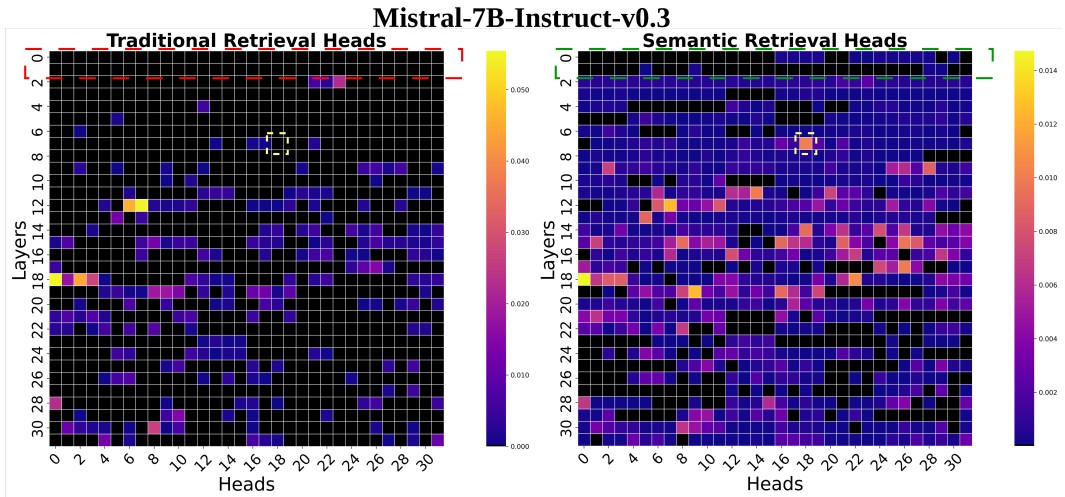

Figure 10: Head visualization for Mistral-7B-Instruct-v0.3. Left: Traditional Retrieval Heads. Right: Semantic Retrieval Heads identified.

## H    DETAILED RESULTS FOR NEEDLE-IN-A-HAYSTACK EVALUATION

This section provides detailed results for the Needle-in-a-Haystack evaluation referenced in the main paper. Figures 11–15 present the corresponding results for the Llama-3.1-8B-Instruct model under KV cache budgets ranging from 128 to 2048. Figures 16–20 present the performance of the Mistral-7B-Instruct-v0.3 model under the same cache budgets. Figures 21–25 present the corresponding results for the Llama-3.1-8B-Instruct model under the same cache budgets. CompressKV consistently achieves the highest accuracy across all settings, demonstrating its superiority over competing compression strategies.

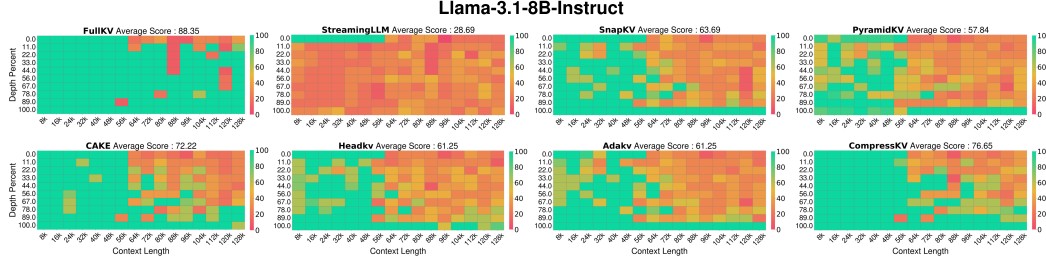

Figure 11: Needle-in-a-Haystack test results on Llama-3.1-8B-Instruct with KV cache = 128.

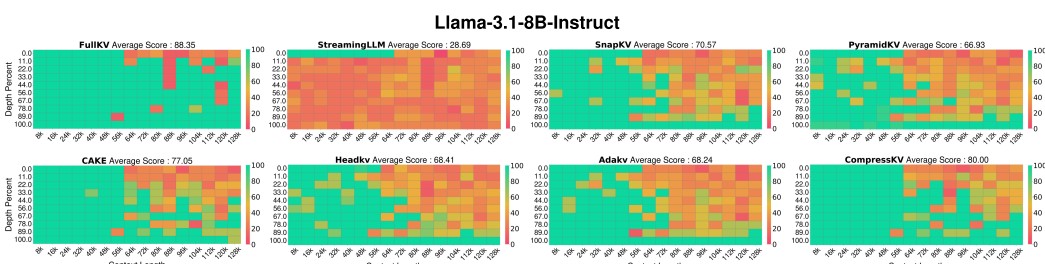

Figure 12: Needle-in-a-Haystack test results on Llama-3.1-8B-Instruct with KV cache = 256.

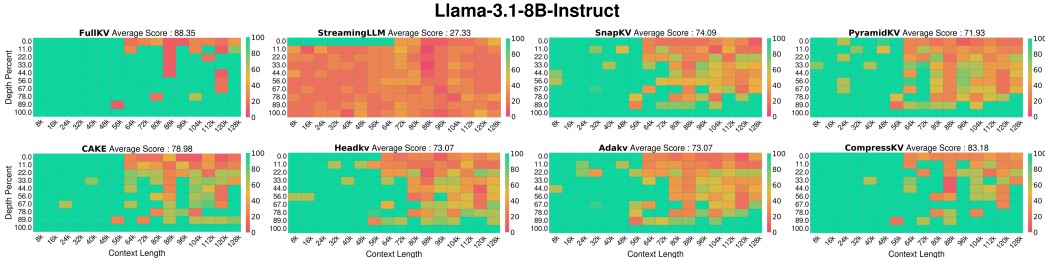

Figure 13: Needle-in-a-Haystack test results on Llama-3.1-8B-Instruct with KV cache = 512.

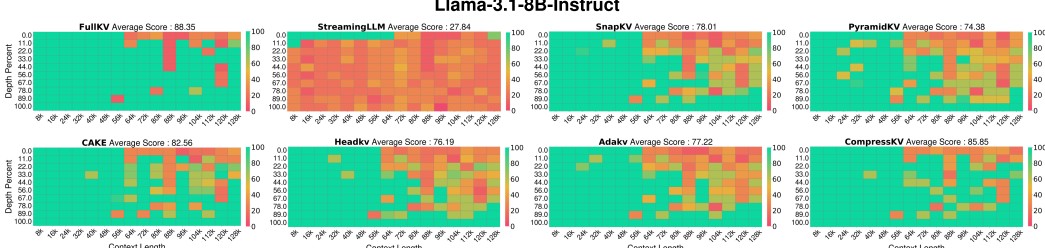

Figure 14: Needle-in-a-Haystack test results on Llama-3.1-8B-Instruct with KV cache = 1024.

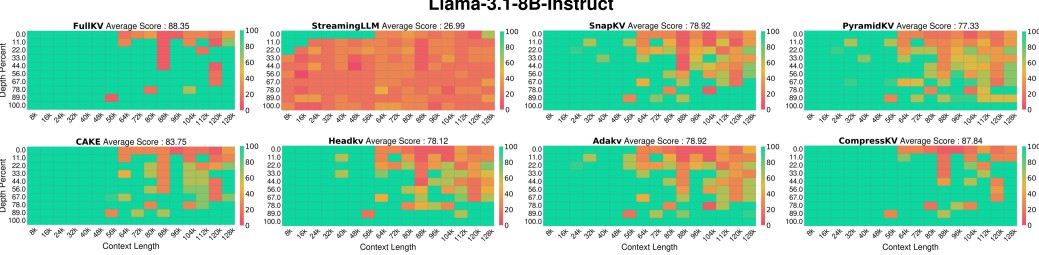

Figure 15: Needle-in-a-Haystack test results on Llama-3.1-8B-Instruct with KV cache = 2048.

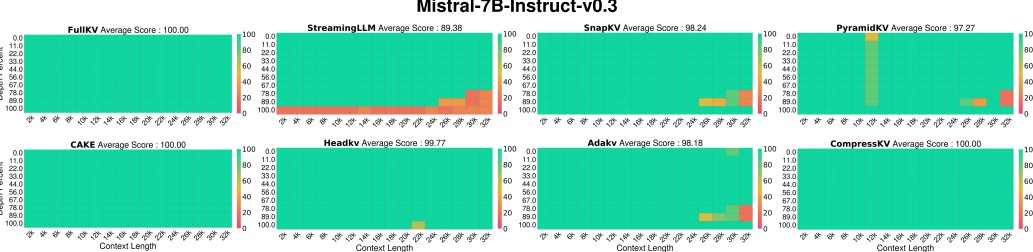

Figure 16: Needle-in-a-Haystack test results on Mistral-7B-Instruct-v0.3 with KV cache = 128.

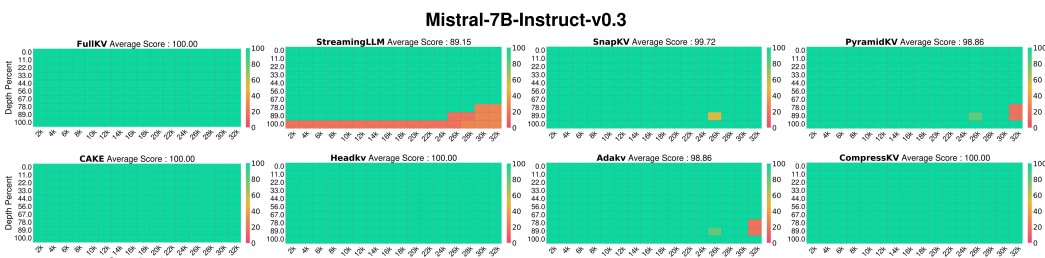

Figure 17: Needle-in-a-Haystack test results on Mistral-7B-Instruct-v0.3 with KV cache = 256.

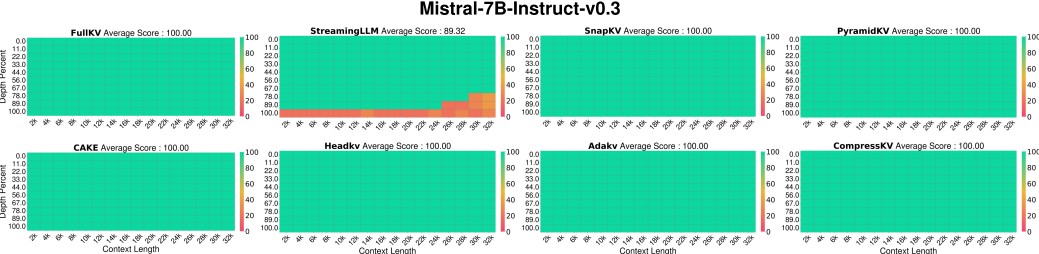

Figure 18: Needle-in-a-Haystack test results on Mistral-7B-Instruct-v0.3 with KV cache = 512.

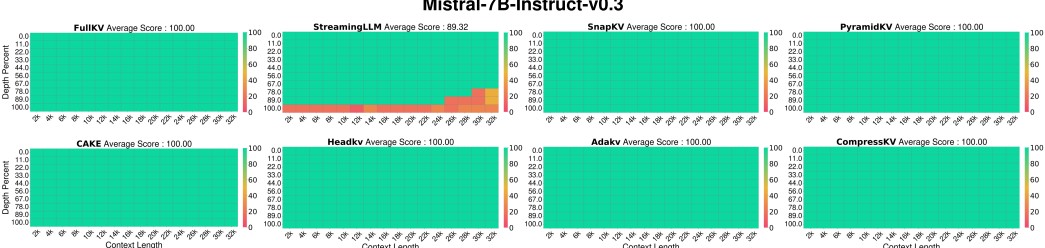

Figure 19: Needle-in-a-Haystack test results on Mistral-7B-Instruct-v0.3 with KV cache = 1024.

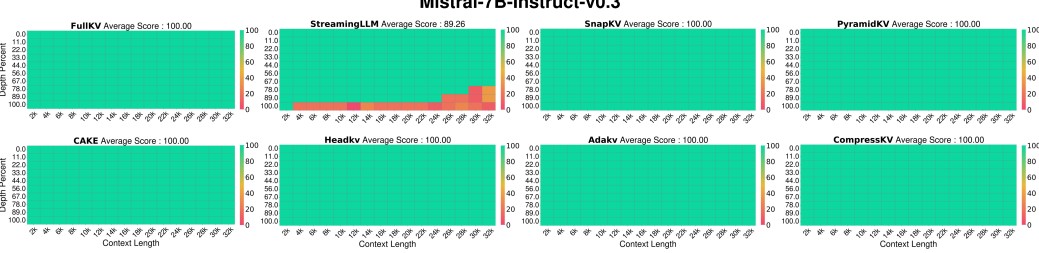

Figure 20: Needle-in-a-Haystack test results on Mistral-7B-Instruct-v0.3 with KV cache = 2048.

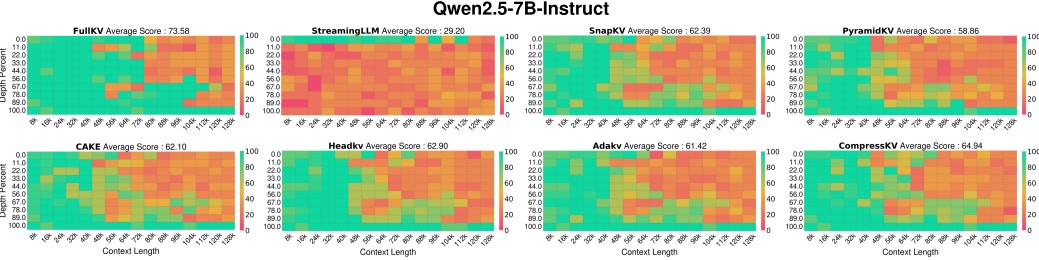

Figure 21: Needle-in-a-Haystack test results on Qwen2.5-7B-Instruct with KV cache = 128.

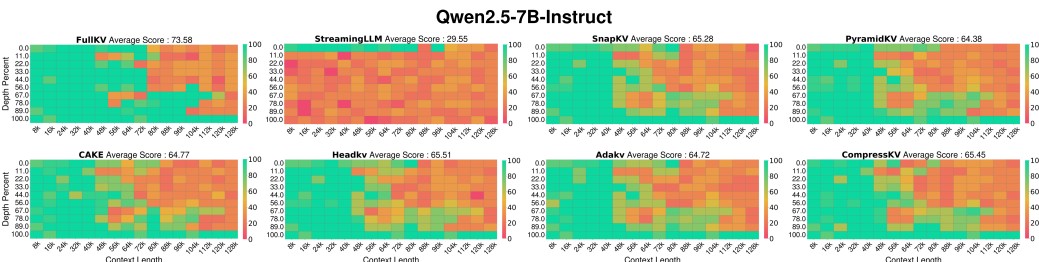

Figure 22: Needle-in-a-Haystack test results on Qwen2.5-7B-Instruct with KV cache = 256.

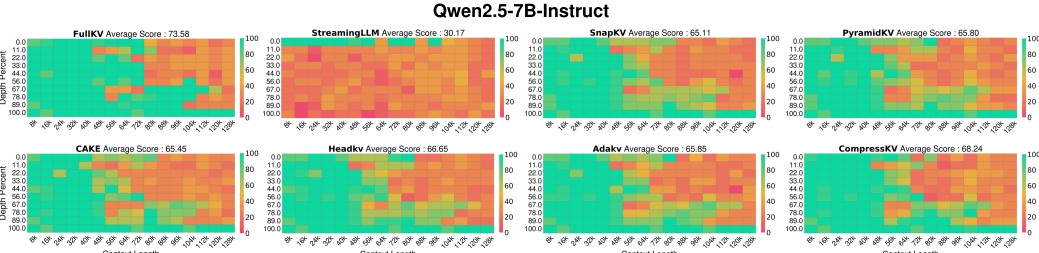

Figure 23: Needle-in-a-Haystack test results on Qwen2.5-7B-Instruct with KV cache = 512.

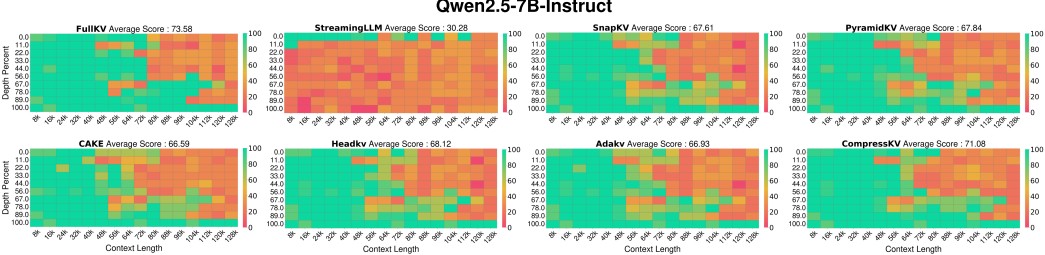

Figure 24: Needle-in-a-Haystack test results on Qwen2.5-7B-Instruct with KV cache = 1024.

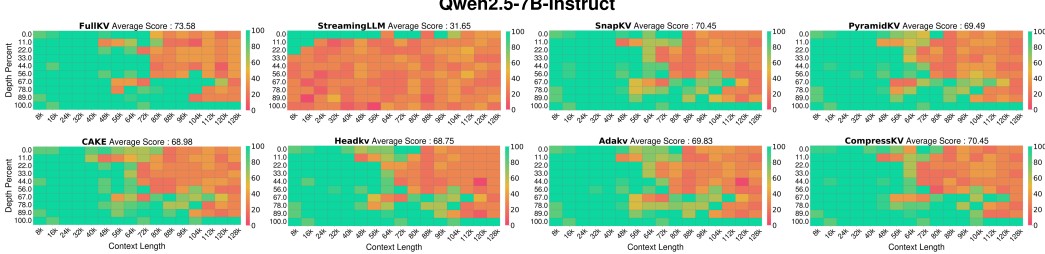

Figure 25: Needle-in-a-Haystack test results on Qwen2.5-7B-Instruct with KV cache = 2048.

# I  COMPREHENSIVE MASKING-BASED ABLATION OF DIFFERENT HEAD TYPES

We extend the masking analysis from the main paper by evaluating the effect of masking the top 10, 20, and 30 Semantic Retrieval Heads and the traditional Retrieval Heads in Mistral-7B-Instruct-v0.3 shown in Figure 26. Our experiments demonstrate that masking the top 30 traditional Retrieval Heads in Mistral-7B-Instruct-v0.3 results in only a $\approx 12\%$ drop in accuracy, whereas masking the top 30 Semantic Retrieval Heads causes a $\approx 74\%$ degradation. These findings underscore the critical role of Semantic Retrieval Heads in overall model performance and validate the superiority of our identification method over conventional head-selection approaches.

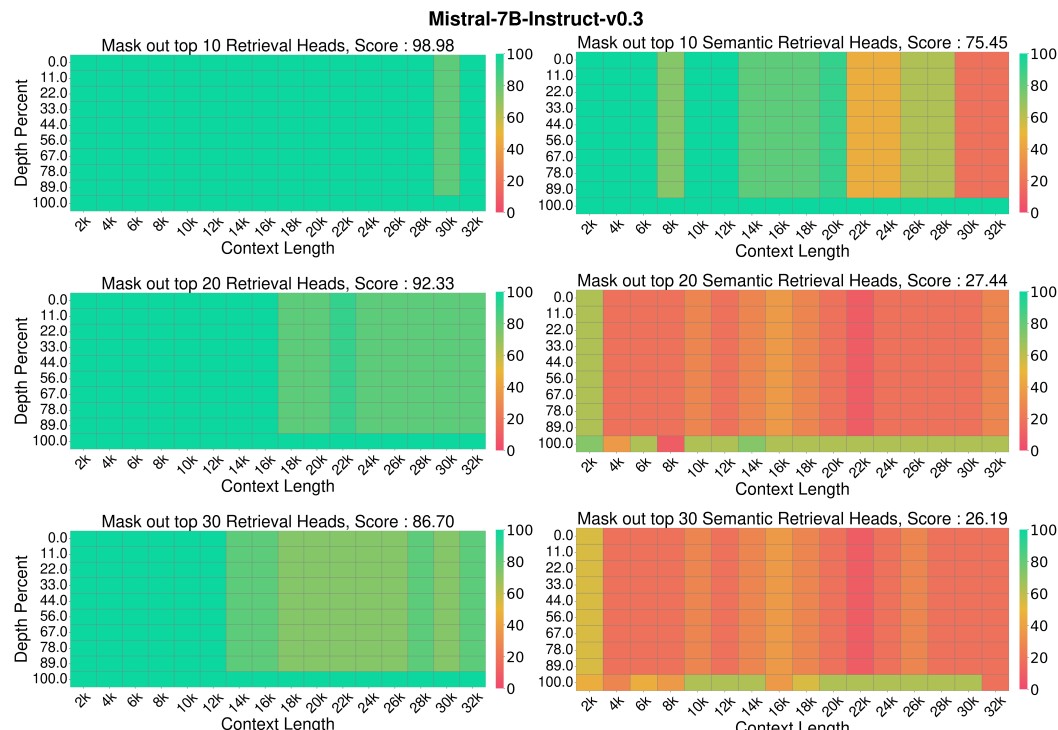

Figure 26: Ablation on the Needle-in-a-Haystack retrieval task for Mistral-7B-Instruct-v0.3. The left column masks the top-k retrieval heads, and the right column masks the top-k semantic retrieval heads. Lower scores indicate heads with the greatest impact on model performance—masking them causes the most severe drop in accuracy.

## J   ORTHOGONAL INTEGRATION WITH QUANTIZAITON

CompressKV is orthogonal to KV-cache quantization approaches such as KIVI (Liu et al., 2024). While KIVI reduces the KV-cache by representing keys and values in low-bit precision (e.g., per-channel/per-token quantization), CompressKV keeps full-precision KV entries but aggressively prunes less critical tokens, thereby shortening the effective sequence length during decoding. As a result, the two methods target different dimensions of the KV-cache bottleneck and can be naturally combined. To verify this orthogonality and clarify their relative strengths, we evaluate (i) CompressKV alone, (ii) KIVI alone, and (iii) the combined CompressKV+KIVI variant on the LongBench benchmark; results are summarized in Table 11. In our main setup, we parameterize compression by the fraction of KV-cache memory saved relative to a 16-bit full-precision baseline. Under this convention, KIVI with 2-bit quantization corresponds to a compression ratio of $0.875$ (i.e., $87.5\%$ memory saving), while an extreme 1-bit setting corresponds to a compression ratio of $0.9375$. At the $0.875$ setting, KIVI attains slightly higher average accuracy than CompressKV at the same overall memory budget. However, when pushing to the more aggressive $0.9375$ setting, KIVI suffers a substantial performance drop, whereas CompressKV remains robust and yields markedly better overall accuracy.

For the combined KIVI_CompressKV variant, we first apply CompressKV to discard approximately $87.5\%$ of the least important tokens and then quantize the remaining KV entries to 2-bit. This yields an overall compression ratio of $0.984375$ (i.e., $98.4375\%$ memory saving relative to a 16-bit full KV cache). As shown in Table 11, this combined scheme maintains accuracy comparable to CompressKV alone and far exceeds pure KIVI at the same extreme compression level, indicating that KIVI and CompressKV are complementary rather than competing techniques.

Table 11: We compare KIVI (quantization), CompressKV (token pruning), and their combination (KIVI+CompressKV) on Llama-3.1-8B-Instruct and Mistral-7B-Instruct-v0.3. "FullKV" denotes the uncompressed 16-bit KV cache. "Compression ratio" denotes the fraction of KV-cache memory saved relative to the 16-bit full-precision baseline (e.g., $0.875$ corresponds to 2-bit quantization, $0.9375$ to 1-bit, and $0.984375$ to applying CompressKV with a token keep ratio of $12.5\%$ followed by 2-bit quantization).

| Method | Single-Document QA | | | Multi-Document QA | | | Summarization | | | Few-shot Learning | | | Synthetic | | Code | | Avg. |
|---|---|---|---|---|---|---|---|---|---|---|---|---|---|---|---|---|---|
| | NrtvQA | Qasper | MF-en | HotpotQA | Musique | 2WikiMQA | GovReport | QMSum | MultiNews | TREC | TriviaQA | SAMSum | PCount | PR-en | Lcc | RB-P | |
| *Llama3.1-8B-Instruct, KV Size = Full* | | | | | | | | | | | | | | | | | |
| FullKV | 30.97 | 45.49 | 53.78 | 54.76 | 31.42 | 47.13 | 34.9 | 25.28 | 27.48 | 73.0 | 91.65 | 43.8 | 6.0 | 99.5 | 63.38 | 56.73 | 49.08 |
| *Llama-3.1-8B-Instruct, Compression ratio = 0.875(2bit)* | | | | | | | | | | | | | | | | | |
| KIVI | 30.85 | 43.95 | 53.97 | 54.38 | 30.34 | 46.11 | 34.48 | 25.33 | 27.16 | 73 | 92.26 | 43.43 | 6.33 | 99 | 62.06 | 55.15 | 48.61 |
| CompressKV | 30.89 | 45.47 | 51.98 | 54.94 | 31.05 | 46.79 | 30.87 | 24.98 | 23.99 | 70.5 | 91.32 | 42.5 | 6 | 99.5 | 62.34 | 56.86 | 48.12 |
| *Llama-3.1-8B-Instruct, Compression ratio = 0.9375(2bit)* | | | | | | | | | | | | | | | | | |
| KIVI | 4.85 | 8.93 | 9.13 | 12.26 | 4.24 | 7.73 | 13.44 | 11.17 | 13.7 | 36.25 | 23.19 | 9.6 | 4.2 | 10.06 | 28.8 | 26.06 | 13.98 |
| CompressKV | 30.77 | 41.48 | 52.17 | 55.25 | 30.81 | 46.81 | 28.21 | 24.05 | 22.16 | 70.5 | 90.93 | 41.37 | 6 | 99.5 | 60.33 | 55.72 | 47.25 |
| *Llama-3.1-8B-Instruct, Compression ratio = 0.984375* | | | | | | | | | | | | | | | | | |
| KIVI+CompressKV | 29.24 | 43.03 | 50.93 | 55.3 | 30.41 | 46.36 | 30.73 | 24.91 | 23.65 | 68.5 | 91.49 | 41.64 | 6.36 | 99.5 | 60.43 | 53.22 | 47.23 |
| *Mistral-7B-Instruct-v0.3, KV Size = Full* | | | | | | | | | | | | | | | | | |
| FullKV | 29.07 | 41.54 | 52.88 | 49.37 | 28.58 | 39.01 | 35.07 | 25.71 | 27.73 | 76.0 | 88.59 | 47.51 | 6.0 | 98.0 | 59.35 | 60.7 | 47.82 |
| *Mistral-7B-Instruct-v0.3, Compression ratio = 0.875(2bit)* | | | | | | | | | | | | | | | | | |
| KIVI | 29.93 | 39.87 | 54.02 | 48.68 | 28.55 | 38.39 | 34.03 | 25.22 | 27.42 | 76 | 88.07 | 47.22 | 5.56 | 95.5 | 58.82 | 59.51 | 47.3 |
| CompressKV | 29.57 | 37.44 | 53.28 | 50.38 | 28.86 | 39.04 | 30.39 | 24.99 | 23.81 | 76 | 89.21 | 45.7 | 5.5 | 97.5 | 59.09 | 60.87 | 46.98 |
| *Mistral-7B-Instruct-v0.3, Compression ratio = 0.9375(2bit)* | | | | | | | | | | | | | | | | | |
| KIVI | 8.61 | 5.57 | 9.68 | 8.37 | 3.72 | 5.99 | 15.45 | 12.51 | 12.16 | 36 | 29.66 | 8.84 | 3.49 | 14.33 | 30.38 | 27.88 | 14.54 |
| CompressKV | 29.14 | 34.81 | 52.42 | 49.55 | 28.85 | 38.1 | 27.95 | 24.76 | 21.94 | 76 | 89.54 | 43.97 | 5.5 | 97 | 57.98 | 60.44 | 46.12 |
| *Mistral-7B-Instruct-v0.3, Compression ratio = 0.984375* | | | | | | | | | | | | | | | | | |
| KIVI+CompressKV | 29.46 | 37.22 | 53 | 48.36 | 27.2 | 38.71 | 29.85 | 24.48 | 23.63 | 76 | 89.42 | 44.54 | 6.06 | 93 | 57.97 | 58.24 | 46.07 |

## K    COMPARISON WITH D2O

We also compare CompressKV against D2O (Wan et al., 2025). D2O is built on H2O-style heavy-hitter estimation and requires explicitly computing the full attention matrix in the prefill stage using the original dense attention kernels. This design is incompatible with FlashAttention-2 (Dao, 2024), and in practice it introduces substantial prefill slowdowns and, at long context lengths, $O(L^2)$ memory pressure that often results in out-of-memory (OOM) errors.

To obtain a practical comparison, we restrict the maximum context length to $8,192$ tokens, and run both methods on Llama-3.1-8B-Instruct on LongBench. Compression ratio denotes the fraction of KV-cache memory saved relative to a 16-bit full-precision KV cache. Thus, keeping a global KV budget (keep ratio) of $0.125$ corresponds to a compression ratio of $1 - 0.125 = 0.875$, and a budget of $0.0625$ corresponds to a compression ratio of $1 - 0.0625 = 0.9375$. Table 12 reports task-wise scores and the average over all LongBench tasks.

At a compression ratio of $0.875$, CompressKV outperforms D2O by $+3.53$ points on average ($44.76$ vs. $41.23$). When we further tighten the budget to $0.0625$ (compression ratio $0.9375$), the gap widens to $+6.20$ points ($43.91$ vs. $37.71$),indicating that token-level compression with CompressKV is more robust under aggressive compression than the heavy-hitter style selection used by D2O.

Table 12: Comparison with D2O on LongBench. We evaluate CompressKV and D2O on Llama-3.1-8B-Instruct with maximum context length 8,192). We report results at two KV compression ratios, following the same convention as for KIVI: compression ratio denotes the fraction of KV-cache memory saved relative to a 16-bit full-precision KV cache. A compression ratio of $0.875$ corresponds to a global KV budget (keep ratio) of $0.125$, and $0.9375$ corresponds to a budget of $0.0625$.

| Method | Single-Document QA | | | Multi-Document QA | | | Summarization | | | Few-shot Learning | | | Synthetic | | Code | | Avg. |
|---|---|---|---|---|---|---|---|---|---|---|---|---|---|---|---|---|---|
| | NrtvQA | Qasper | MF-en | HotpotQA | Musique | 2WikiMQA | GovReport | QMSum | MultiNews | TREC | TriviaQA | SAMSum | PCount | PR-en | Lcc | RB-P | |
| Llama-3.1-8B-Instruct, KV compression ratio = 0.875 | | | | | | | | | | | | | | | | | |
| CompressKV | 25.16 | 44.16 | 52.30 | 46.56 | 24.73 | 45.72 | 29.16 | 22.94 | 23.87 | 67.5 | 91.51 | 44.10 | 5.95 | 76.0 | 62.16 | 54.28 | 44.76 |
| D2O | 25.52 | 32.54 | 41.42 | 45.68 | 22.36 | 42.61 | 26.71 | 22.78 | 23.32 | 58.5 | 90.40 | 41.25 | 5.75 | 74.5 | 55.85 | 50.54 | 41.23 |
| Llama-3.1-8B-Instruct, KV compression ratio = 0.9375 | | | | | | | | | | | | | | | | | |
| CompressKV | 25.17 | 41.19 | 51.18 | 47.16 | 24.12 | 45.80 | 27.25 | 22.87 | 21.92 | 67.0 | 90.85 | 43.05 | 7.22 | 76.0 | 59.92 | 51.81 | 43.91 |
| D2O | 23.56 | 26.02 | 36.07 | 43.48 | 20.91 | 39.48 | 23.67 | 21.92 | 21.62 | 51.0 | 89.01 | 38.22 | 6.55 | 74.5 | 43.62 | 43.76 | 37.71 |

## L ADDITIONAL ABLATION ON LAYER-WISE BUDGET ALLOCATION AND ADAPTIVE NUMBER OF SRH

### L.1 ABLATION ON LAYER-WISE BUDGET ALLOCATION

Table 13: **Additional ablation on layer-wise budget allocation.** We report LongBench average accuracy (%) of Mistral-7B-Instruct-v0.3 with a fixed KV budget of 256 tokens per layer, under different choices of dataset $\mathcal{D}$ used to compute layer importance scores in Eq. (9)–(12). Results are nearly identical, indicating that our layer-wise allocation is robust to the choice of $\mathcal{D}$.

| Layer-score source | Acc. (%) |
|---|---|
| All-tasks Average | 45.43 |
| NarrativeQA-only | 45.45 |
| Qasper-only | 45.43 |
| QMSum-only | 45.38 |
| TriviaQA-only | 45.36 |

Here, we perform an additional ablation in which the layer scores are derived from a single dataset rather than from the average across all tasks.

Concretely, we consider Mistral-7B-Instruct-v0.3 on LongBench with a fixed KV budget of 256 tokens per layer (the same setting as in the main ablation in Table 3). We compute layer importance scores in five ways: (i) averaging reconstruction errors over all LongBench datasets (our default), and (ii) using only NarrativeQA, (iii) only Qasper, (iv) only QMSum, or (v) only TriviaQA as the dataset $\mathcal{D}$ for Eq. (9)–(12). In all cases, the resulting layer scores are normalized and used to allocate the same global KV budget across layers.

Table 13 reports task-wise scores and averages. The overall performance is virtually identical across all variants: the average accuracy ranges from 45.36% to 45.45%, and the differences with respect to the "All-tasks Average" profile are less than $0.08$ points. This indicates that our layer-wise budget allocation is robust to the choice of dataset used to compute layer importance and does not overfit to a particular task; averaging over all tasks is a convenient and stable default.

### L.2 ADAPTIVE-$k$ SRH SELECTION

Here, we further evaluate an adaptive-$k$ variant. For each layer, we first L1-normalize the SRH scores and then select all heads whose normalized score exceeds a threshold $\tau$, while enforcing a minimum of 4 heads per layer. We sweep $\tau \in \{0.001, 0.002, 0.003, 0.004, 0.006, 0.007\}$ on

Mistral-7B-Instruct-v0.3 with a KV budget of 256 tokens per layer. As summarized in Table 14, all adaptive-$k$ configurations achieve average LongBench accuracy that is very close to the fixed top-4 baseline and do not exhibit a consistent improvement trend. This suggests that a small fixed $k$ is already a stable and effective design choice, and we therefore adopt top-4 SRH per layer for all main experiments for simplicity and reproducibility.

Table 14: Ablation on adaptive-$k$ SRH selection (Mistral-7B-Instruct-v0.3, KV budget = 256). We compare fixed top-4 SRH per layer with adaptive-$k$ variants using different thresholds $\tau$ on L1-normalized SRH scores (with a minimum of 4 heads per layer).

| Method | Acc. (%) |
|---|---|
| Fixed top-4 SRH | 44.96 |
| Adaptive-$k$ ($\tau = 0.001$) | 44.84 |
| Adaptive-$k$ ($\tau = 0.002$) | 44.74 |
| Adaptive-$k$ ($\tau = 0.003$) | 44.75 |
| Adaptive-$k$ ($\tau = 0.004$) | 44.80 |
| Adaptive-$k$ ($\tau = 0.005$) | 44.89 |
| Adaptive-$k$ ($\tau = 0.006$) | 44.87 |
| Adaptive-$k$ ($\tau = 0.007$) | 44.88 |

## M  THROUGHPUT

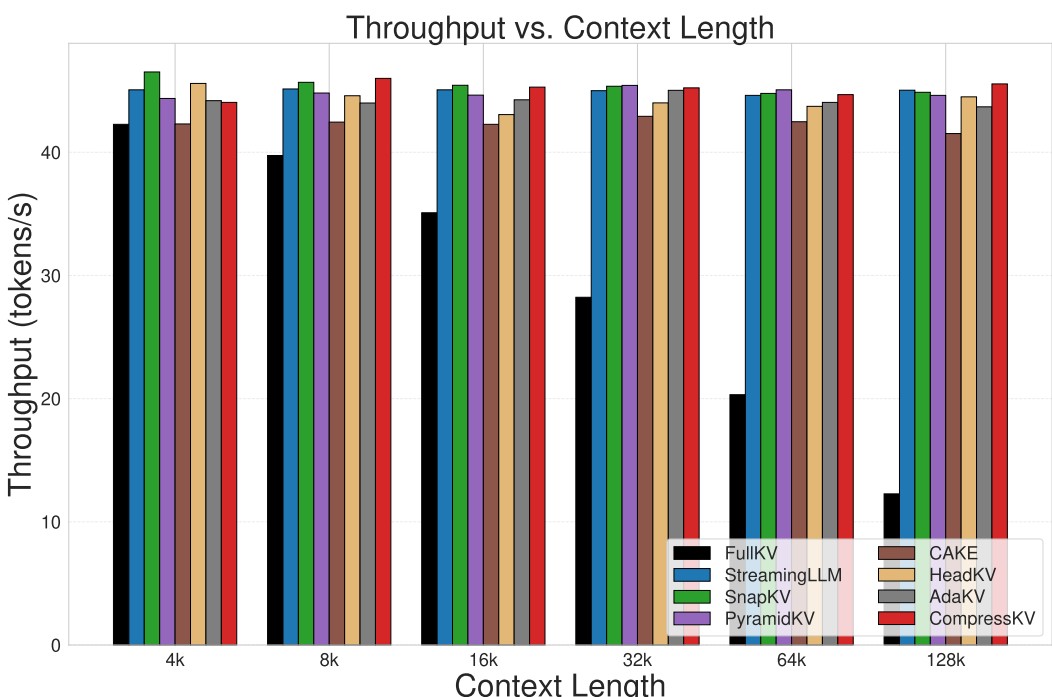

Figure 27: Decoding throughput vs. context length. Throughput (tokens/s) of Llama-3.1-8B-Instruct evaluated on a single NVIDIA A100 GPU as the context length increases from 4k to 128k under different KV-cache eviction approaches (higher is better).

Figure 27 shows the decoding throughput (tokens/s) of `Llama-3.1-8B-Instruct` as a function of context length from 4k to 128k under different KV-cache management strategies. All the setting are same with the Section 4.4 As expected, the FullKV baseline is the slowest configuration and its throughput degrades noticeably as the context grows, illustrating the quadratic cost of attending over an ever-expanding KV cache. In contrast, all KV compression and streaming-based methods (StreamingLLM, SnapKV, PyramidKV, CAKE, HeadKV-GQA, AdaKV-GQA, and CompressKV)

maintain a much flatter throughput curve and consistently outperform FullKV across all context lengths.

# N   EVALUATION ON LONGBENCH V2

We evaluate Llama-3.1-8B-Instruct on LongBench V2(QA Tasks) using the official evaluation protocol. For all methods, we adopt chain-of-thought prompting ("cot" in Table 15). With a full 128k KV cache, the FullKV baseline attains an overall accuracy of 35.8. At a 0.75 KV Compression ratio, StreamingLLM suffers a large degradation (22.7 overall), whereas eviction-based methods such as SnapKV, PyramidKV, CAKE, HeadKV-GQA, AdaKV-GQA, and CompressKV substantially close the gap to FullKV. Among these, SnapKV and CompressKV achieve the strongest overall performance (32.2 and 32.0, respectively). At a more aggressive 0.9375 KV Compression ratio, the differences between methods become more pronounced. StreamingLLM drops to 18.1 overall, while all cache-compression approaches remain in the 27–32% range. CompressKV reaches 32.2 overall accuracy, outperforming SnapKV (31.2), PyramidKV (30.0), CAKE (27.2), HeadKV-GQA (30.2), and AdaKV-GQA (28.2).

Table 15: Results on LongBench V2 benchmark with Llama-3.1-8B-Instruct. All results use chain-of-thought (CoT) prompting. We report overall accuracy and breakdowns by difficulty (Easy/Hard) and context length (Short/Medium/Long). "KV compression ratio" denotes the fraction of the original KV cache that is removed. All numbers are accuracies in %.

| Method | Overall Acc. | Easy | Hard | Short | Medium | Long |
|--------|--------------|------|------|-------|--------|------|
| Llama-3.1-8B-Instruct, Full KV cache | | | | | | |
| FullKV | 35.8 | 42.7 | 31.5 | 40.6 | 32.6 | 34.3 |
| KV Compression ratio = 0.75 (25% of full KV cache) | | | | | | |
| StreamingLLM | 22.7 | 22.9 | 22.5 | 32.2 | 19.5 | 13.0 |
| SnapKV | 32.2 | 37.5 | 28.9 | 32.8 | 31.6 | 32.4 |
| PyramidKV | 30.6 | 34.9 | 28.0 | 31.1 | 30.2 | 30.6 |
| CAKE | 30.4 | 33.3 | 28.6 | 30.6 | 30.7 | 29.6 |
| HeadKV-GQA | 30.0 | 30.7 | 29.6 | 34.4 | 28.4 | 25.9 |
| AdaKV-GQA | 30.4 | 34.4 | 28.0 | 34.4 | 29.8 | 25.0 |
| CompressKV | **32.0** | 35.4 | **29.9** | **35.6** | **32.6** | 25.0 |
| KV Compression ratio = 0.9375 (6.25% of full KV cache) | | | | | | |
| StreamingLLM | 18.1 | 21.4 | 16.1 | 22.2 | 17.2 | 13.0 |
| SnapKV | 31.2 | 33.3 | 29.9 | 37.2 | 28.4 | 26.9 |
| PyramidKV | 30.0 | 30.7 | 29.6 | 34.4 | 27.0 | 28.7 |
| CAKE | 27.2 | 28.6 | 26.4 | 30.0 | 23.3 | 30.6 |
| HeadKV-GQA | 30.2 | 33.3 | 28.3 | 34.4 | 28.4 | 26.9 |
| AdaKV-GQA | 28.2 | 25.0 | 30.2 | 35.0 | 24.7 | 24.1 |
| CompressKV | **32.2** | 33.3 | **31.5** | 35.0 | **34.0** | 24.1 |

## O   COMPRESSKV PSEUDO CODE

Following SnapKV Li et al. (2024), we use the most recent **window_size** tokens as the observation window to analyze how different local contexts influence attention allocation patterns. In all experiments, we fix **window_size** = 8. **kernel_size** denotes the size of the 1D pooling window used to smooth local fluctuations in the attention weights before ranking tokens; we set **kernel_size** = 5 in our experiments. **layer_budget** denotes the per-layer KV-cache capacity. As shown in List 1, CompressKV is smoothly compatible with FlashAttention-family kernels: we only compress the KV cache during the prefill stage, while subsequent attention computations simply invoke the standard FlashAttention2 kernel on the compressed KV cache, without any modification to the kernel itself.

Listing 1: Implementation of CompressKV in pseudo PyTorch style.

```python
def compress_kv_update_kv(
    self, key_states, query_states, value_states, num_key_value_groups, layer_idx,)
    bsz, num_heads, q_len, head_dim = query_states.shape
    num_kv_heads = key_states.shape[1]
    # do not compress if under budget
    layer_budget = self.max_capacity_prompt_layer_adaptive[layer_idx]
    if q_len < layer_budget:
        return key_states, value_states
    # gather important heads for this layer
    important_head_cl = torch.tensor(self.important_heads[layer_idx],    )
    # project to the KV heads corresponding to important query heads
    key_states_imp = get_important_head_kv(key_states, num_key_value_groups, important_head_cl
        )
    # local attention window over the last window_size tokens
    q_win = query_states[:, important_head_cl, -self.window_size:, :]
    attn_weights = compute_attn(q_win, key_states_imp, attn_masks).sum(dim=-2)
    # Apply 1D pooling for clustering
    attn_cache = pool1d(attn_weights, kernel_size=kernel_size, padding=kernel_size//2
        , stride=1).mean(dim=1)
    # select top-k historical tokens according to pooled scores
    budget = layer_budget - self.window_size
    indices = attn_cache.topk(budget, dim=-1).indices  # [bsz, budget]
    # expand indices to full KV heads and head_dim
    indices = indices.unsqueeze(1).unsqueeze(-1)        # [bsz, 1, budget, 1]
    indices = indices.expand(-1, num_kv_heads, -1, head_dim)
    # compress past part of the KV cache
    k_past = key_states[:, :, :-self.window_size, :]
    v_past = value_states[:, :, :-self.window_size, :]
    k_past_compress = k_past.gather(dim=2, index=indices)
    v_past_compress = v_past.gather(dim=2, index=indices)
    # always keep the most recent window_size tokens
    k_cur = key_states[:, :, -self.window_size:, :]
    v_cur = value_states[:, :, -self.window_size:, :]
    key_states = torch.cat([k_past_compress, k_cur], dim=2)
    value_states = torch.cat([v_past_compress, v_cur], dim=2)
    return key_states, value_states

def attn_forward(self, hidden_states, ....., past_key_value: Optional[Cache] = None):

    bsz, q_len, _ = hidden_states.size()
    ...
    # update KV cache with CompressKV during prefill
    if past_key_value is not None:
        if prefill
            key_states_compress, value_states_compress = compress_kv_update_kv(
                key_states, query_states, value_states, self.num_key_value_groups, self.layer_idx,
            )
            past_key_value.update(key_states_compress, value_states_compress, self.layer_idx,...
            )
        else:  # decoding step, just append new token
            key_states, value_states = past_key_value.update(key_states, value_states, self.
                layer_idx, cache_kwargs
            )

    # core FlashAttention2 call
    attn_output = _flash_attention_forward(query_states, key_states, value_states, ...)
    attn_output = attn_output.reshape(bsz, q_len, -1).contiguous()
    attn_output = self.o_proj(attn_output)
    return attn_output, past_key_value
```

