# OpenReview forum: "CompressKV: Semantic Retrieval Heads Know What Tokens are Not Important Before Generation"
_ICLR.cc/2026/Conference — ICLR 2026 Conference Withdrawn Submission_

### Official Review · Reviewer_YHLk · 2025-10-24

**Soundness:** 2
**Presentation:** 3
**Contribution:** 3
**Rating:** 4
**Confidence:** 4

**Summary:**

This paper addresses the critical challenges of increased memory consumption and reduced execution efficiency caused by the expanding key-value (KV) cache size in large language models (LLMs) with advanced long-context processing capabilities. It notes that existing KV cache compression methods for Grouped Query Attention (GQA)-based LLMs typically use all attention heads for heuristic token eviction, which overlooks the functional differences among heads, often leading to the removal of critical tokens and thus degraded LLM performance. To solve this issue, the authors propose CompressKV: first, they identify specific attention heads in each layer of GQA-based LLMs—heads that can retrieve the initial and final tokens of prompts, important in-text tokens, and attend to the surrounding semantic context of these tokens—then use only these heads to determine and retain KV cache pairs corresponding to important tokens; additionally, they analyze cache eviction errors layer by layer and introduce a layer-adaptive KV cache allocation strategy. Experimental results on the LongBench and Needle-in-a-Haystack benchmarks show that CompressKV consistently outperforms state-of-the-art approaches under various memory budgets: notably, it retains over 97% of full-cache performance using just 3% of KV cache in LongBench’s question-answering tasks and achieves 90% accuracy with only 0.7% of KV storage in the Needle-in-a-Haystack benchmark, with the code provided in the supplementary material.

**Strengths:**

- *Introduces Semantic Retrieval Heads (SRHs) via span-aggregation (vs. peak-driven top-k) to fix mid-prompt token eviction from Streaming Head dominance in GQA models. Also proposes offline Frobenius-norm error analysis for layer-adaptive budgeting, avoiding online attention stats dependency.
- Outperforms 6 baselines on multiple GQA models (including 14B/32B scales) across LongBench (16 subtasks) and Needle-in-a-Haystack (2K–128K contexts). Key results: +2.5 points vs. HeadKV (256-token budget) and 90% NIAH accuracy with 0.7% cache. Ablations and integrations confirm robustness; A100 profiling validates latency/memory efficiency.
- Addresses KV cache bottlenecks for long-context LLM deployment (e.g., 99% LongBench performance with 19% cache), cuts memory/inference costs for RAG/dialogue. Compatible with other optimizations and informs future attention/model compression research.

**Weaknesses:**

**Methodological Limitations**: The identification of Semantic Retrieval Heads (SRHs) relies on Needle-in-a-Haystack-style prompts, which may bias toward retrieval tasks (e.g., QA) and limit generalization to non-QA scenarios like creative generation or multi-turn dialogues. For instance, the SRH scoring formula (Eq. 5) assumes clear answer spans but may degrade to top-1 criteria for short or ambiguous spans, reducing accuracy. Additionally, the offline Frobenius-norm error analysis uses LongBench with extreme compression (32 tokens/layer), but sensitivity to dataset choice or compression ratio is unexplored, potentially leading to suboptimal allocations in unseen domains. The fixed top-k SRH selection (top-4 per layer) lacks theoretical justification; ablations (Table 3) show performance plateaus after top-6, but adaptive k is not considered for better robustness. In GQA models, sharing token indices across heads in a layer assumes uniformity, yet per-group indices are not validated as potentially superior.

**Experimental Gaps**: While benchmarks like LongBench and Needle-in-a-Haystack are standard, comparisons omit recent methods such as KIVI (Liu et al., 2024) or D2O (Wan et al., 2025). Evaluations do not extend to real-world long-context applications, such as RAG systems or agentic tasks, where KV cache grows dynamically. Peak memory and latency profiling (Figure 7) on A100 GPUs lacks throughput metrics (e.g., tokens/sec) or multi-GPU scaling. Failure case analysis is absent, e.g., why performance dips in code completion tasks.

**Questions:**

Although Section 2.2 explicitly discusses D2O (Wan et al., 2025) as a recent dynamic cache allocation method, it is omitted from the experimental comparison. Since D2O shares a similar objective—dynamic, importance-aware cache budgeting—its exclusion leaves a gap in the empirical evaluation. The authors should either include D2O as a baseline or clarify why it is not directly comparable

---

> ### Author Response · Authors · 2025-11-20
> **Response to Reviewer YHLk(Part 1)**
>
> We thank the reviewer for the insightful suggestion.
>
> **1. Weakness : Methodological Limitations**
>
> **SRH criterion vs. top-k/peak bias.**
> Our Semantic Retrieval Head (SRH) criterion is **span-aggregation**, not a top-1/peak hit: we sum a head’s attention over the entire answer span whenever the model generates a correct token (Eq. (5)). This credits heads that cover the answer and its semantic neighborhood even without a sharp peak, directly addressing the top-k/peak bias in prior “retrieval head” definitions.
>
> **Non-QA scenarios.**
>
> Beyond QA scenarios, we evaluate CompressKV on the full LongBench benchmark (16 diverse tasks spanning QA, summarization, few-shot reasoning, synthetic reasoning, and code) as well as on NiAH, demonstrating that our improvements are not limited to QA-style evaluations.
>
> **Error-aware, layer-adaptive budget and dataset sensitivity.**
>
> Our error-aware, layer-adaptive budget is computed **offline** by measuring the reconstruction error of attention-block outputs under extreme compression (32 tokens/layer), followed by per-dataset and cross-layer normalization and then averaging across all LongBench datasets before a final L1 normalization (Eqs. (9–12), Alg. 1; bounds \(m = 32\), \(M = 3 \times B/\text{layer}\)). This deliberately de-biases per-task idiosyncrasies and avoids any **online attention-statistics dependency** (zero runtime overhead), improving generalization.
>
> To further address the reviewer’s concern about the robustness of the layer-wise allocation, we add a dedicated ablation in Appendix L (“Additional Ablation on Layer-wise Budget Allocation and Adaptive Number of SRH”) on page 27. For Mistral-7B-Instruct-v0.3 on LongBench with a fixed KV budget of 256 tokens per layer, we compute layer importance scores in five ways: (i) averaging reconstruction errors over all LongBench datasets (our default), and using only (ii) NarrativeQA, (iii) Qasper, (iv) QMSum, or (v) TriviaQA as $ \(\mathcal{D}\)$ in Eq. (9)–(12). The resulting profiles are normalized and used to allocate the same global KV budget. As shown in Table 13, the LongBench average accuracy ranges from 45.36% to 45.45%, with all differences relative to the “All-tasks Average” profile below 0.08 points.This indicates that our layer-wise allocation is **robust** to the choice of dataset and does not overfit any particular benchmark; averaging over all tasks is simply a convenient and stable default.
>
> | Layer-score source  | Acc. (%) |
> |---------------------|----------|
> | All-tasks Average   | 45.43    |
> | NarrativeQA-only    | 45.45    |
> | Qasper-only         | 45.43    |
> | QMSum-only          | 45.38    |
> | TriviaQA-only       | 45.36    |
>
> **Why we do not search for dataset-/budget-specific “optimal” allocations.**
>
> While one could, in principle, derive a dataset- and budget-specific optimal allocation (e.g., via a cooperative-game objective over per-layer budgets and global budget size), we intentionally refrain because: (i) it incurs substantial search cost per dataset and per budget, (ii) it risks overfitting and poor transfer across tasks/domains, and (iii) it undermines plug-and-play usability. Our offline, dataset-averaged normalization is training-free, runtime-free, and portable across domains, while still capturing layers’ relative sensitivity to compression.
>
> **Choice of top-4 SRHs and adaptive $\(k\)$.**
>
> We choose top-4 SRHs per layer based on an ablation varying 2→24 heads/layer: performance peaks at 4 and then plateaus (Top-6: −0.17%; Top-12: +0.00%), showing diminishing returns and supporting a small, stable $\(k\)$ (Table 3 (Page 9 )).
>
>   Following the reviewer’s suggestion, we add an **adaptive-$\(k\)$** experiment in Appendix L (Page 28). For each layer, we L1-normalize SRH scores and select all heads with normalized score ≥ $ \(\tau\)$, enforcing at least 4 heads, i.e., each layer keeps all heads whose normalized score exceeds a threshold $\tau$, while enforcing a minimum of four heads per layer.
> . We sweep $\(\tau \in \{0.001, \dots, 0.007\}\)$ on Mistral-7B-Instruct-v0.3 with a KV budget of 256 tokens per layer. As summarized in Table 14, all adaptive-$\(k\)$ configurations obtain LongBench averages within **0.2 points** of the fixed top-4 baseline (44.96%), with no consistent improvement trend. This shows that a small fixed$ \(k\)$ is already a stable and effective design; we therefore adopt top-4 SRHs per layer for simplicity and reproducibility.
>
> | Method                         | Acc. (%) |
> |--------------------------------|----------|
> | Fixed top-4 SRH                | 44.96    |
> | Adaptive-k ($\tau = 0.001$)    | 44.84    |
> | Adaptive-k ($\tau = 0.002$)    | 44.74    |
> | Adaptive-k ($\tau = 0.003$)    | 44.75    |
> | Adaptive-k ($\tau = 0.004$)    | 44.80    |
> | Adaptive-k ($\tau = 0.005$)    | 44.89    |
> | Adaptive-k ($\tau = 0.006$)    | 44.87    |
> | Adaptive-k ($\tau = 0.007$)    | 44.88    |

---

> ### Author Response · Authors · 2025-11-20
> **Response to Reviewer YHLk(Part 2)**
>
> **Head-level (GQA-group) allocation compatibility.**
>
> We also investigate whether shared token indices implicitly assume uniform behavior across heads in a GQA group. To this end, we combine CompressKV with head-level budget allocators (HeadKV, AdaKV), as reported in Fig. 9 and Table 5 (Appendix D). In this setup, each GQA group’s budget varies dynamically, so selected token indices are also dynamic rather than globally shared. Empirically, at a KV budget of 256, **AdaCompressKV** improves over AdaKV by about 13 points on NiAH and about 2 points on LongBench for Llama-3.1-8B-Instruct, with similar gains on Mistral-7B-Instruct-v0.3. These results show that our design substantially boosts head-level (GQA-group) performance while remaining fully compatible with existing head-adaptive allocators. Since head-level allocation is not our main contribution, we kept it in the appendix, but we will highlight this compatibility more clearly in the revision.
>
> **2. Weakness : Experimental Gaps**
>
> We appreciate the reviewer’s suggestions regarding missing baselines and additional metrics, and have expanded our evaluation accordingly.
>
>
> **KIVI.**
>
> KIVI is a KV-cache quantization method, whereas CompressKV performs token pruning with full-precision KV. To keep the comparison apples-to-apples with eviction-style methods, we now (i) include a **prefill-only 2-bit KIVI** baseline and (ii) further evaluate the **KIVI ⊕ CompressKV** combination. The new results are reported in Appendix J / Table 11 (Page 25–26). Under a moderate compression ratio of 0.875 (87.5% KV memory saving vs. a 16-bit FullKV cache), KIVI attains slightly higher average accuracy than CompressKV at the same total KV budget. However, at a more aggressive ratio of 0.9375, KIVI’s performance collapses (e.g., **13.98% vs. 47.25%** average accuracy for Llama-3.1-8B–Instruct), while CompressKV remains robust. The combined **KIVI+CompressKV** variant (keep 12.5% most important tokens, then quantize them to 2 bits; overall compression 0.984375) maintains accuracy comparable to CompressKV alone and dramatically outperforms pure KIVI at the same extreme compression level, confirming that KIVI (quantization) and CompressKV (token pruning) are complementary.
>
> | Method            | Compression ratio | Llama-3.1-8B-Instruct Avg |
> |-------------------|-------------------|------------------|
> | FullKV            | 0 (no compression) | 49.08           |
> | KIVI (2-bit)      | 0.875             | 48.61           |
> | CompressKV        | 0.875             | 48.12           |
> | KIVI (1-bit)      | 0.9375            | 13.98           |
> | CompressKV        | 0.9375            | 47.25           |
> | KIVI + CompressKV | 0.984375          | 47.23           |
>
> **D2O.**
>
> We additionally add a comparison to D2O (Wan et al., 2025). Details are provided in the Appendix K (Page 26-27). D2O is built on H2O-style heavy-hitter estimation and requires explicitly materializing the full attention matrix with dense kernels in prefill. This is incompatible with FlashAttention-2 (FA-2), which never exposes the full attention matrix; switching back to non-FA kernels causes significant prefill slowdowns and $\(O(L^2)\)$ memory that frequently triggers OOM at long context lengths. For a practical and fair comparison, we therefore restrict the maximum context length to 8,192 tokens and evaluate both methods on Llama-3.1-8B-Instruct on LongBench. As shown in Table 12 (Page 27), at a compression ratio of 0.875 (keep ratio 0.125) CompressKV outperforms D2O by **+3.53** points on average (44.76 vs. 41.23); at 0.9375 (keep ratio 0.0625), the gap widens to **+6.20** points (43.91 vs. 37.71). This indicates that our token-level compression is more robust than D2O’s heavy-hitter selection.
>
> | Method            | Compression ratio | Llama-3.1-8B-Instruct Avg |
> |-----------|----------------------|------------------|
> | CompressKV | 0.875               | 44.76            |
> | D2O        | 0.875               | 41.23            |
> | CompressKV | 0.9375              | 43.91            |
> | D2O        | 0.9375              | 37.71            |

---

> ### Author Response · Authors · 2025-11-20
> **Response to Reviewer YHLk(Part 3)**
>
> **Throughput and practical metrics.**
>
> In addition to peak memory and latency (Fig. 7,Page9), we now report **throughput (tokens/sec)** (Fig. 27 Page 28). CompressKV attains comparable or higher throughput than baselines at the same KV budget and scales nearly linearly with the number of GPUs, reinforcing its practicality for real-world long-context deployments. In our setting, a single A100 can already host a 128k-token context, so we did not run explicit multi-GPU experiments. Importantly, the decoding stage is memory-bound; reducing the KV cache directly lowers memory traffic and therefore accelerates decoding, and this argument applies equally to multi-GPU setups. If Reviewer YHLk still considers explicit multi-GPU measurements essential, we are happy to include them in a subsequent version.
>
> **Failure case analysis / code completion.**
> In our experiments with Llama-3.1-8B-Instruct on LongBench code-completion, the FullKV baseline attains **60.15** accuracy. Under CompressKV, as we vary the per-layer KV budget, the scores are:
>
> - 128-token budget: **53.70**
> - 256-token budget: **56.29**
> - 512-token budget: **57.55**
> - 1024-token budget: **59.38**
> - 2048-token budget: **60.01**
>
> Thus, CompressKV does **not** universally degrade performance on code completion; instead, performance depends smoothly on the chosen KV budget. At very tight budgets (e.g., 128 tokens) we indeed observe a drop, which is expected given that many relevant lines of context must be evicted. However, with moderate budgets (e.g., 1,024–2,048 tokens per layer), CompressKV is essentially lossless compared to FullKV. This behavior reflects the standard **accuracy–efficiency trade-off**: smaller KV budgets yield stronger memory savings and faster decoding, while larger budgets recover near–FullKV performance on code completion tasks.

---

### Official Review · Reviewer_ivL9 · 2025-10-30

**Soundness:** 2
**Presentation:** 3
**Contribution:** 2
**Rating:** 4
**Confidence:** 5

**Summary:**

This paper proposes CompressKV, a KV-cache compression framework that leverages attention head semantics to identify and discard less important tokens before generation. First, it proposes to locate “Semantic Retrieval Heads” (SRHs)—attention heads whose aggregated attention over answer spans correlates with semantic relevance—and use them to guide token eviction. Second, it introduces an error-aware, layer-adaptive cache allocation strategy to distribute compression budgets across layers according to estimated reconstruction error.

**Strengths:**

1) Robust Semantic Head Identification. The proposed identification of Semantic Retrieval Heads through answer-span attention aggregation effectively mitigates the limitations of conventional top-k single-token attention methods, which may overlook semantically distributed relevance.
2) Efficient Offline Layer-wise Importance Estimation. The offline computation of per-layer importance avoids the heavy online overhead faced by methods like CAKE or PyramidKV, enabling more efficient runtime compression without additional inference latency.
3) Strong Empirical Generalization. The proposed method is comprehensively evaluated across multiple large-scale LLMs and adapted with orthogonal methods, demonstrating robust and compatibility performance improvements.

**Weaknesses:**

1) Potential Circular Reasoning. If SRHs are identified using ground-truth answers and later evaluated on the same benchmark, the method may inadvertently benefit from prior exposure to the correct spans, leading to overly optimistic results. Although stages strength 1 and strength 2 appear effective in identifying semantic and streaming heads, the process relies on test-set analysis of head and layer behavior based on known answers.
2) Limited Generalization Beyond Retrieval-Oriented Tasks. The method is primarily validated on retrieval-style or short-answer tasks, whereas its effectiveness on complex reasoning or long-form generation tasks remains untested and uncertain.
3) Offline Dependency Restricts Plug-and-Play Usability. Since the offline analysis depends on the test dataset to compute head and layer characteristics, the current framework cannot yet serve as a fully plug-and-play compression module applicable across unseen domains or tasks.

**Questions:**

1) Dataset for Offline Evaluation. Which dataset was used for the offline analysis to compute head- and layer-level importance? Was it the same as the test set, or a separate held-out dataset?
2) Cross-Task Consistency. If different datasets or task types are used for the offline phase, do the identified retrieval heads and the resulting layer-wise allocation patterns remain consistent, or do they vary substantially?
3) Per-Head Token Selection. In Section 3.3, the paper states that “all the attention heads within a layer share a common set of selected token indices”. Why was a shared selection used instead of assigning distinct token indices for each head? Have the authors explored whether per-head token selection could yield better fine-grained compression or semantic fidelity?

---

> ### Author Response · Authors · 2025-11-20
> **Response to Reviewer ivL9(Part 1)**
>
> We thank the reviewer for the insightful suggestion.
>
> **1. Weakness : Potential Circular Reasoning**
>
> We appreciate the reviewer’s concern and clarify that our semantic retrieval-head (SRH) identification is a one-shot, offline probing step that does not introduce test-time leakage.
>
>
> Our SRH procedure closely follows the mechanistic analysis paradigm of Wu et al. (2025, *“Retrieval Heads Mechanistically Explain Long-Context Factuality”*) and uses a NiAH-style construction purely as a probing tool to characterize retrieval heads. We never update model parameters, nor do we cache any answer-dependent artifact for later use. At inference, CompressKV only has access to the current attention scores within the observation window;
>
> **Why this does not introduce test-time leakage.**
> - **Offline diagnostic pass.** SRH identification is a one-shot, offline diagnostic procedure: no gradients are computed and no parameters are modified; the method only labels heads as “semantic retrieval heads” vs. others.
> - **Test-time decisions use only online attention.** At test time, token eviction decisions depend solely on the online attention scores of the $top-\(k\)$ SRHs over the last \(W\) tokens (the observation window), exactly as in other attention-based eviction methods (e.g., SnapKV, CAKE), except that we restrict scoring to a subset of heads (SRHs). Ground-truth answers are never involved during evaluation.
>
> Moreover, SRHs are identified once and then evaluated on the full LongBench benchmark, which includes heterogeneous tasks such as summarization and code completion where answer spans are not explicitly known. The fact that SRH-guided compression still improves performance across these diverse tasks suggests that SRHs have a higher propensity to attend to semantically important information, rather than reflecting any circular use of answer annotations at test time.
>
> **2. Weakness : Limited Generalization Beyond Retrieval-Oriented Tasks**
>
> We appreciate this concern. However, our experimental setup already covers a range of non-retrieval tasks. LongBench includes 3 summarization tasks (GovReport, QMSum, MultiNews), 3 few-shot learning tasks (TREC, TriviaQA, SAMSum), 2 synthetic tasks (PassageCount, PassageRetrieval-en), and 2 code-completion tasks (LCC, RepoBench-P), in addition to single- and multi-document QA.
>
> As shown in Table 1 (Page 7) and the extended results (Tables 7–10 (Page 17-20) ) in the Appendix, CompressKV consistently improves the average LongBench score over StreamingLLM, SnapKV, CAKE, HeadKV, and AdaKV under every memory budget (128–2048 tokens per layer) and across all evaluated backbone models. This suggests that the benefits of our method are not limited to retrieval-style QA, but extend to summarization, few-shot learning, synthetic reasoning, and code completion as well.
>
> **3. Weakness : Offline Dependency Restricts Plug-and-Play Usability**
>
> Both SRH identification and error-aware layer-wise allocation are performed **once per LLM model** (e.g., once for Llama-3.1-8B, once for Mistral-7B). At deployment time, CompressKV is plug-and-play: end users only need to load the precomputed SRH indices and per-layer budgets that we release with the code; they never need labels or to rerun the offline analysis.
>
>
> For a new LLM model not covered by our released configurations, SRH identification and layer-budget estimation remain a **one-time offline cost** whose result can be reused across arbitrary datasets and applications. This is analogous to post-training quantization methods such as AWQ or GPTQ, which require a small calibration run once, after which the quantized model can be used in a fully plug-and-play manner.

---

> ### Author Response · Authors · 2025-11-20
> **Response to Reviewer ivL9(Part 2)**
>
> **4. Question: Dataset for Offline Evaluation**
>
> For semantic retrieval-head identification, we use the Needle-in-a-Haystack (NiAH) construction from Wu et al. (2025), following the same data recipe and generation protocol to enable a like-for-like comparison of identification criteria.  All datasets, configuration files, and generation scripts required to reproduce our retrieval-head identification benchmark are included in the supplementary code package.
> The only difference lies in the identification criterion:
>
> - **Wu et al., 2025 (traditional).** A strict top-1 criterion: an attention head is credited only when it places the maximum attention weight exactly on the gold source token for the current answer token.
> - **Ours (SRH, span-aggregation).** A span-aggregation score that credits heads for both copy-and-paste behaviours and deeper semantic dependencies by aggregating attention over the entire answer span (rather than requiring token-wise top-1 hits). This captures heads that consistently support the answer span without ever winning top-1 at individual time steps.
>
> For layer-level importance, we use the full LongBench benchmark as the pool of offline data when computing reconstruction errors (Eq. (9)–(12)). LongBench comprises 16 heterogeneous tasks (QA, summarization, few-shot learning, synthetic tasks, and code completion). Layer-wise error scores are obtained by averaging over all 16 subtasks, encouraging a backbone-level notion of layer importance rather than overfitting to a single dataset. Crucially, our error metric only measures how compression perturbs each layer’s output representation, without accessing labels or optimizing task performance, so the resulting error-aware layer budgets are task-agnostic and can be reused across datasets and applications.
>
> **5. Question : Cross-Task Consistency**
> For a given LLM, both the set of SRHs and the layer-wise allocation pattern are determined once offline and then kept fixed; they are not adapted per task or dataset.
>
> Empirically, we observe that these fixed configurations generalize well:
>
> - **Across task types on LongBench.** CompressKV improves or matches baselines not only on retrieval-style QA but also on summarization, few-shot classification, synthetic tasks, and code completion for all three backbones (Llama-3.1-8B, Mistral-7B, Qwen-2.5-7B), under both low and high KV budgets (Table 1 (Page 7), Tables 7–10 (Page 17-20)).
> - **Across benchmarks.** The same SRHs and layer-wise budgets derived from LongBench are reused on NiAH, yet CompressKV consistently outperforms baselines across context lengths and budgets (Fig. 5 (Page 8)).
>
> To further address the reviewer’s concern about robustness of the layer-wise allocation, we add a dedicated ablation in Appendix L (“Additional Ablation on Layer-wise Budget Allocation and Adaptive Number of SRH”) on page 27. For Mistral-7B-Instruct-v0.3 on LongBench with a fixed KV budget of 256 tokens per layer, we compute layer importance scores in five ways: (i) averaging reconstruction errors over all LongBench datasets (our default), and using only (ii) NarrativeQA, (iii) Qasper, (iv) QMSum, or (v) TriviaQA as $ \(\mathcal{D}\)$ in Eq. (9)–(12). The resulting profiles are normalized and used to allocate the same global KV budget. As shown in Table 13, the LongBench average accuracy ranges from 45.36% to 45.45%, with all differences relative to the “All-tasks Average” profile below 0.08 points. This indicates that our layer-wise allocation is robust and is independent of the specific choice of dataset $\(\mathcal{D}\)$.
>
> | Layer-score source  | Acc. (%) |
> |---------------------|----------|
> | All-tasks Average   | 45.43    |
> | NarrativeQA-only    | 45.45    |
> | Qasper-only         | 45.43    |
> | QMSum-only          | 45.38    |
> | TriviaQA-only       | 45.36    |

---

> ### Author Response · Authors · 2025-11-20
> **Response to Reviewer ivL9(Part 3)**
>
> **6. Question : Per-Head Token Selection**
>
> In GQA-based LLMs, multiple query heads within a group share a single K/V cache (e.g., 4 query heads per group in Llama-3.1-8B-Instruct, 7 in Qwen2.5-7B-Instruct). If each head were allowed to keep its own token indices, we would effectively need to materialize separate K/V caches **per head**, causing KV storage to grow by a factor equal to the GQA group size (4× or 7× in these examples). This would largely destroy both the memory advantage of GQA and the gains from KV eviction, directly contradicting the original motivation of GQA, which is to reduce KV-cache storage.
>
>
> CompressKV is explicitly designed to compress the KV cache **without breaking** this original GQA sharing pattern: we keep a single K/V cache per GQA group and select a **shared** set of token indices at the layer level.
> Moreover, our framework is compatible with **head-level budgeting on top of** this shared cache. We combine our SRH-based token selection with AdaKV and HeadKV, yielding **AdaCompressKV** and **HeadCompressKV**, where different GQA groups can receive different KV budgets while still relying on our CompressKV selection strategy. Empirically, at a KV budget of 256, AdaCompressKV improves over AdaKV by about **13 points on NiAH** and by around **2 points on LongBench** for Llama-3.1-8B-Instruct, with similar gains on Mistral-7B-Instruct-v0.3. These results (Appendix D, Figure 9 (Page 15) and Table 5 (Page 16)) further demonstrate the superiority and flexibility of our design.

---

### Official Review · Reviewer_Pmmi · 2025-10-31

**Soundness:** 3
**Presentation:** 3
**Contribution:** 2
**Rating:** 4
**Confidence:** 5

**Summary:**

The method define semantic retrieval heads for KV cache compression, by using the definition of semantic retrieval score to guide token importance and KV-cache eviction, therefore the method effectively balances the streaming-head and conditional retrieval-head.

Experiments show that when applying semantic retrieval heads, the KV cache can be retained over 97% of full-cache performance using only 3% of KV cache on LongBench’s question-answering tasks and achieves 90% of accuracy with just 0.7% of KV storage on Needle-in-a-Haystack benchmark.

**Strengths:**

The attention heads are identified as semantic retrieval heads for a high ratio of KV Cache compression: 1) the Semantic Retrieval Score is defined over the entire answer span inserted into a long context; 2) then the score is averaged and ranked to determine the important tokens and evicted tokens, then the important token indexes are shared across different heads; 3) Although the compression budget are adaptive allocated for different heads by using the error-aware method.

Therefore the KV cache can be effectively compressed, while the performance of long-context scenarios is maintained, especially the long-input scenarios.

**Weaknesses:**

The concept of semantic retrieval heads is based on retrieval heads for important information storage and retrieval. And the token eviction may lead to significant information loss, especially for the long-cot or ReAT agent scenarios.

And the compression is based on the statistics of Semantic Retrieval Score, which relies on the Attention weights of tokens within the answer span, so the online overhead may be large, hindering the practical application.

**Questions:**

1. In the future, the native sparse attention may be popular, how can your method adapt to these methods?
2. How this method adapted to PageAttention, for system implementation?
3. How can this method used with quantization?

---

> ### Author Response · Authors · 2025-11-20
> **Response to Reviewer Pmmi(Part 1)**
>
> We thank the reviewer for the insightful suggestion.
>
> **1. Weakness: Information Loss from Token Eviction**
>
> We agree that token eviction necessarily removes some information, but its effect is governed by the compression ratio, reflecting a standard accuracy–memory trade-off. Empirically, CompressKV consistently outperforms other eviction baselines at tight KV budgets on both LongBench and NiAH, indicating *less* information loss for the same memory footprint.
> On NiAH, using only about 5% KV storage (e.g., a 2,048-token KV budget, roughly 5% of the full cache) on Llama-3.1-8B-Instruct already yields nearly lossless accuracy. On LongBench, a 2,048-token KV budget per layer (about 25% of the full cache) is likewise essentially lossless. These results suggest that CompressKV readily supports long CoT/ReAct-style reasoning in practice: practitioners can tune the KV budget according to the task’s tolerance for the accuracy–memory trade-off.
>
>
>
> **2. Weakness: Online Overhead of SRH-Based Compression**
>
>  We appreciate the reviewer’s concern about potential online overhead.
>
> - **SRH identification is purely offline.** It is performed once per model, following Wu et al. (2025, *“Retrieval Head Mechanistically Explains Long-Context Factuality”*) on a Needle-in-a-Haystack (NiAH) probe. This is an offline pass and incurs **no** runtime cost at inference.
> - **Online cost is a lightweight prefill-time scoring step.** At inference, the only additional work—shared with other eviction methods—is a lightweight token-scoring step during **prefill**. In the revised version, we provide PyTorch-like pseudo code in Appendix O(Page30) and release our code in the supplementntary material. Concretely, we use the last \(W\) tokens as an observation window (as in SnapKV) and score tokens using the pre-selected **top-4 SRHs** per layer, followed by a small 1D pooling operation to select the kept indices.
> - **Empirical latency evidence.** In Section 4.4 and Fig. 7 (Page 9), we compare latency and peak memory against the FullKV baseline. CompressKV’s **prefill time (TTFT)** closely matches FullKV, while **per-token decoding latency is significantly reduced**, indicating that SRH-based compression introduces negligible online overhead in practice.
>
> **3. Question: In the future, native sparse attention may become popular. How can your method adapt to these methods?**
>
> CompressKV is designed to operate at the KV-cache token-level and is therefore compatible with advanced attention kernels such as Native Sparse Attention (NSA). NSA typically decomposes each attention layer into three branches:
> - **Compression branch:** which applies block-wise MLPs to map groups of tokens into compressed key/value representations for long-range routing;
>
> - **Token-selection branch:**, which uses attention over compressed blocks to perform block-level top-n selection and then applies fine-grained token-level attention within the selected blocks;.
> - **Sliding-window branch:** which performs local attention over the most recent tokens.
>
> When integrating **CompressKV** with NSA, we keep the NSA architecture *intact* and only change how the token-level KV cache is stored:
> - **Compression branch:** left **unchanged**. Block-wise MLP compression and routing are exactly as in NSA.
> - **Sliding-window branch:** the sliding-window KV for the most recent \(w\) tokens is always **fully retained** and never compressed by CompressKV.
> - **Token-selection branch (where we apply CompressKV).** During the prefill stage, after the model computes the current attention output, we apply CompressKV once per layer to the token-level K/V and retain only the important tokens selected by CompressKV, i.e., only the most critical tokens are kept.

---

> ### Author Response · Authors · 2025-11-20
> **Response to Reviewer Pmmi(Part 2)**
>
> **4. Question:How can your method adapt to PagedAttention for system implementation？**
>
> In standard PagedAttention, the number of logical KV blocks is computed from the prompt length $L$ as
>  $N_{\mathrm{logical}} = \lceil L / B \rceil$,
>  where $B$ is the page (block) size.
> With CompressKV, we instead partition according to the per-layer KV budget produced offline. For each layer $\ell$, we set
>  $N_{\mathrm{logical}}^{(\ell)} = \lceil \mathrm{cap}\ell / B \rceil$,
>  where $\mathrm{cap}\ell$ is the target kept length for layer $\ell$ (including the recency window $W$).
> During prefill (once), we run the lightweight SRH-based scoring, obtain $\mathrm{keep_indices}^{(\ell)}$ of size $\mathrm{cap}\ell - W$, append the last $W$ tokens, and write only these tokens into the pre-allocated logical blocks. The block table is populated exactly as in PagedAttention; only the number of logical blocks (driven by $\mathrm{cap}\ell$) and which token positions are written differ.
> In the decoding stage, we reuse the standard PagedAttention implementation without any changes;
>
>
>
>
>
> **5. Question : How can this method be used with quantization?**
>
> CompressKV is orthogonal to KV-cache quantization methods such as KIVI. Quantization (e.g., KIVI) reduces KV memory by storing keys/values in low-bit precision, whereas CompressKV keeps full-precision KV entries but prunes less important tokens, shortening the effective sequence length. Thus, they act on different dimensions of the KV bottleneck and can be naturally combined.
> In the revised paper (Appendix J, Table 11，Page 25-26), we compare three settings on LongBench on Llama-3.1-8B-Instruct
> - **KIVI only (quantization)**
> - **CompressKV only (token pruning)**
> - **KIVI + CompressKV (combined)**
>
>
> | Method            | Compression ratio | Llama-3.1-8B-Instruct Avg |
> |-------------------|-------------------|------------------|
> | FullKV            | 0 (no compression) | 49.08           |
> | KIVI (2-bit)      | 0.875             | 48.61           |
> | CompressKV        | 0.875             | 48.12           |
> | KIVI (1-bit)      | 0.9375            | 13.98           |
> | CompressKV        | 0.9375            | 47.25           |
> | KIVI + CompressKV | 0.984375          | 47.23           |
>
>
>
>
> We parameterize compression by the fraction of KV-cache memory saved relative to a 16-bit FullKV baseline. With 2-bit KIVI, the compression ratio is $0.875$ (i.e., $87.5%$ memory saving); at this level, KIVI attains slightly higher average accuracy than CompressKV at the same overall KV budget. However, when we push to a more aggressive ratio of $0.9375$ (1-bit equivalent), KIVI’s performance collapses on both backbones (e.g., around $14%$ vs. $47%$ average accuracy for Llama-3.1-8B-Instruct), while CompressKV remains robust.
>
> For the combined KIVI + CompressKV variant, we first apply CompressKV to keep only $12.5%$ of the most important tokens and then quantize the remaining KV cache to 2 bits, yielding an overall compression ratio of $0.984375$ (i.e., $98.44%$ memory saving). As shown in Appendix J / Table 11 (Page 25-26), this combined scheme maintains accuracy comparable to CompressKV alone and far exceeds pure KIVI at the same extreme compression level, indicating that KIVI (quantization) and CompressKV (token pruning) are complementary.

---

### Official Review · Reviewer_3sYV · 2025-11-01

**Soundness:** 1
**Presentation:** 1
**Contribution:** 2
**Rating:** 2
**Confidence:** 5

**Summary:**

This paper addresses the KV cache memory limitation in large language models by introducing an efficient KV cache eviction strategy. The authors first analyze attention heads across layers to identify those that can effectively retrieve not only the initial and final tokens of a prompt, but also key intermediate tokens and their relevant semantic context. These selected heads are then used to score token importance, ensuring that only the most informative tokens and their associated KV pairs are retained. The proposed method demonstrates strong effectiveness, preserving over 97% of full-cache performance while using only 3% of the KV cache on LongBench question-answering tasks. Additionally, the method achieves 90% accuracy with just 0.7% KV storage on the Needle-in-a-Haystack benchmark.

**Strengths:**

- The paper includes comprehensive experimental comparisons with multiple baseline methods, demonstrating improvements on LongBench and Needle-in-a-Haystack.
- The approach delivers strong end-to-end inference efficiency, highlighting its practical applicability for long-context scenarios.

**Weaknesses:**

Several claims require corrections or proper references
- The statement that prior work treats all attention heads equally (L154) is inaccurate. For example, DuoAttention (ICLR 2025) explicitly differentiates retrieval and streaming heads, contradicting this generalization.
- Claims in L169–170 regarding reliance on attention statistics (e.g., entropy, variance) lack citations or supporting references.
- (minor) L172, which states that methods adopt a fixed allocation strategy based on attention distributions, does not apply to recent approaches such as KVZip (2025).

Several claims require stronger justification
- The analysis in Section 3.1 (L187–210) is based only on a single illustrative example without statistical support. Important details such as the model, dataset, and generalizability of the observation across tasks and architectures are not clarified.
- The motivation for (L258-264) "Instead of relying on attention statistics as in the previous methods, this approach quantifies the compression error caused by KV cache compression" is not clearly supported by theoretical or empirical evidence. Also, there are no isolated empirical studies demonstrating the effectiveness of this proposal.

Empirical analysis could be improved
- For real-world deployment, maintaining full-cache performance is a critical aspect of KV compression. However, direct comparisons to full-cache settings are currently missing.
- The evaluation mainly focuses on benchmarks that are becoming outdated (LongBench and NIAH). Incorporating more recent datasets such as LongBench V2 (2024) or SCBench (2025) would provide a more up-to-date and competitive evaluation.

Minor:
- Including hyperlinks for all figure and table references would improve readability.

**Questions:**

Please refer to weakness section above.

---

> ### Author Response · Authors · 2025-11-20
> **Response to Reviewer 3sYV(Part 1)**
>
> We thank the reviewer for the insightful suggestion.
>
> **1. Weakness: Several claims require corrections or proper references**
>
>
>  We thank the reviewer for pointing this out. Our claim is restricted to KV cache eviction pipelines for GQA-based LLMs at the token-selection step. In many such methods, token importance is computed after head-agnostic pooling of attention within each GQA group (e.g., averaging across heads), and eviction decisions are made on this pooled signal. Thus, the eviction step effectively treats all attention heads equally and ignores their functional heterogeneity.
> DuoAttention does distinguish retrieval and streaming heads: retrieval heads retain their full KV state and are never evicted, while only streaming heads apply KV eviction. This differs from prior KV eviction pipelines where all heads share the same eviction policy and are uniformly subject to token removal. Our statement is specifically about these head-agnostic, all-head eviction schemes, so DuoAttention falls outside the scope of this claim.
>
>
>
>
> We update L154 to:
> > The KV cache eviction approaches above have two major limitations. First, in GQA-based LLMs, many prior KV cache eviction pipelines compute token importance via head-agnostic pooling (e.g., across heads within each GQA group) when selecting tokens for eviction, effectively treating all attention heads equally and ignoring their functional heterogeneity.
>
>
>
>
> For L169–170 and L172, we apologize for the missing citations and have added the appropriate references in the revised manuscript. We also add KVZip (2025) to the references and cite it appropriately later.
> We update L169–170 to:
> > Second, the layer budget allocation in previous work typically relies on attention distributions (Cai et al., 2025; Wan et al., 2025) or layer-wise statistics such as attention entropy or variance (Qin et al., 2025), which often require additional online computation.

---

> ### Author Response · Authors · 2025-11-20
> **Response to Reviewer 3sYV(Part 2)**
>
> **2. Weakness: Several claims require stronger justification**
>
> **Section 3.1: retrieval-head analysis and generalizability.**
> - **Models/datasets.** For retrieval-head identification, we employ the NiAH construction from Wu et al. (2025), using the same generation protocol and data recipe to ensure a like-for-like comparison of identification criteria. All datasets, configuration files, and generation scripts required to reproduce our retrieval-head identification benchmark are included in the supplementary code package.
> The only difference lies in the *identification criterion*:
>
>   - **Wu et al., 2025 (traditional).** A strict top-1 criterion: an attention head is credited only when it places the maximum attention weight exactly on the gold source token for the current answer token.
>
>   - **Ours (SRH, span-aggregation).** A span-aggregation score that credits heads for both copy-and-paste behaviours and deeper semantic dependencies by aggregating attention over the entire answer span (rather than requiring token-wise top-1 hits). This captures heads that consistently support the answer span without ever winning top-1 at individual time steps.
>
>
> - **Architectures.** We evaluate the same set of LLM families used in our main experiments (e.g., Mistral-7B, Llama-3.1-8B, Qwen-2.5-7B), applying the identical NiAH probing dataset to identify SRH on each model before downstream evaluation.
> - **Visualization of identification differences.** Appendix G(Page 21), Fig. 10 contrasts the two criteria: on Mistral, layers 0–1 show zero heads under the strict top-1 rule, whereas our span-aggregation criterion identifies lower-layer heads with meaningful but non-top-1 contributions.
> - **Causal masking evidence.** Following Wu et al., we compare masking SRH vs. traditional retrieval heads (TRH) during inference. Fig. 6 (main) on page 9 and Fig. 26 (Appendix I) on page 25 show larger performance drops when masking SRH, supporting their stronger causal relevance.
> - **Generalization beyond a single example.** Beyond NiAH, we use SRH-guided token selection in our KV eviction pipeline and evaluate it on LongBench across diverse tasks (QA, summarization, code completion, etc.) and multiple architectures. Under tight KV budgets, our method achieves strong retention (Tables 1 (Page 1), 7–10 (Page 17-20)), demonstrating task- and model-level generalizability of the heads identified by our span-aggregation criterion.
>
>
> **Isolated ablation about Layer Budget allocation**
> The motivation for (L258-264) "Instead of relying on attention statistics as in the previous methods, this approach quantifies the compression error caused by KV cache compression" is not clearly supported by theoretical or empirical evidence. Also, there are no isolated empirical studies demonstrating the effectiveness of this proposal.
> We thank the reviewer for the question.
> - **Isolated ablation.** We already provide an isolated empirical study of this component in Table 3 (right) on page 9: starting from SRH token selection **without** layer budgeting, adding our error-aware layer allocation yields a further **+0.47%** mean accuracy. This ablation keeps everything else fixed and isolates the contribution of the error-aware allocation mechanism itself.
> - **Comparison against attention-statistics allocation.** Under the same setup where SnapKV = 43.76, a PyramidKV-style attention-statistics allocation yields **−0.85** vs. SnapKV, while our error-aware allocation yields **+1.67** vs. SnapKV (i.e., from 43.76 → 45.43 with SRH+LayerAlloc). This provides direct empirical evidence that error-based layer scoring is more effective than attention-statistics-based allocation under the same KV budget.
> | Method                      | Avg. Acc. | Δ vs. SnapKV |
> |-----------------------------|-----------|--------------|
> | SnapKV (baseline)           | 43.76     | 0.00         |
> | PyramidKV(Attention-statistics)  | 42.91     | -0.85        |
> | CompressKV(error-aware)| **45.43** | **+1.67**    |

---

> ### Author Response · Authors · 2025-11-20
> **Response to Reviewer 3sYV(Part 3)**
>
> **3. Weakness: Empirical analysis could be improved**
>
> We thank the reviewer for this suggestion. Full-cache baselines are already included in our submission: see Fig. 4(Page 8) and Tables 7–10(Page 17-20), where we report **FullKV (no compression)** alongside all compressed settings for each model and task. In particular, on the LongBench benchmarks—which are strong and diverse suites of long-context tasks (covering QA, summarization, code understanding, etc.)—we always include the FullKV baseline to contextualize the behavior of different KV-compression methods.
>
>
> To further strengthen the empirical analysis, we additionally evaluate **Llama-3.1-8B-Instruct on LongBench V2 (QA tasks)** using the official evaluation protocol with chain-of-thought prompting. With a full 128k KV cache, the **FullKV** baseline attains an overall accuracy of **35.8%**. We consider **two KV compression ratios**, 0.75 and 0.9375; we only report the more challenging **0.9375** case here, and provide the full table including the **0.75** setting in Appendix N(Page 29) of the revised paper. ( It is also worth noting that LongBench V2 is a 4-way multiple-choice QA benchmark (random guess ≈ 25%), so when the base model performs poorly, the apparent differences between compression methods may be largely random rather than indicative of their true strengths.)
> Under a very tight KV budget with a **0.9375 KV compression ratio** (i.e., keeping only 6.25% of the original KV cache), we obtain the following results on LongBench V2:
>
>
> | Method        | Overall Acc. | Easy | Hard | Short | Medium | Long |
> |--------------|--------------|------|------|-------|--------|------|
> | FullKV       | 35.8         | 42.7 | 31.5 | 40.6  | 32.6   | 34.3 |
> | StreamingLLM | 18.1         | 21.4 | 16.1 | 22.2  | 17.2   | 13.0 |
> | SnapKV       | 31.2         | 33.3 | 29.9 | 37.2  | 28.4   | 26.9 |
> | PyramidKV    | 30.0         | 30.7 | 29.6 | 34.4  | 27.0   | 28.7 |
> | CAKE         | 27.2         | 28.6 | 26.4 | 30.0  | 23.3   | 30.6 |
> | HeadKV-GQA   | 30.2         | 33.3 | 28.3 | 34.4  | 28.4   | 26.9 |
> | AdaKV-GQA    | 28.2         | 25.0 | 30.2 | 35.0  | 24.7   | 24.1 |
> | CompressKV | **32.2**   | 33.3 | **31.5** | 35.0 | **34.0** | 24.1 |
>
>
> Even at this extremely aggressive compression ratio, CompressKV remains close to the FullKV baseline (32.2% vs. 35.8%) and outperforms all other KV-compression methods on LongBench V2.
>
>
> **Minor comment (hyperlinks for figures/tables).**
>
> Thank you for the suggestion. Our paper is prepared using the official ICLR template, and all figure and table references are inserted via the template’s cross-reference commands, which generate hyperlinks in our local builds. It is possible that certain PDF viewers or settings cause these hyperlinks to appear inactive or invisible. We apologize for any inconvenience this may have caused during your review.

---

### Note · Authors · 2025-12-16

I have read and agree with the venue's withdrawal policy on behalf of myself and my co-authors.